# A Multi-Objective Ensemble Approach to Hydrological Modelling in the UK: An Application to Historic Drought Reconstruction

Katie A. Smith[1], Lucy J. Barker[1], Maliko Tanguy[1], Simon Parry[1], Shaun Harrigan[2], Tim P. Legg[3], Christel Prudhomme[2,1,4], and Jamie Hannaford[1,5]

[1] Centre for Ecology & Hydrology, Maclean Building, Benson Lane, Crowmarsh Gifford, Wallingford, Oxfordshire, OX10 8BB, UK
[2] European Centre for Medium-Range Weather Forecasts, Shinfield Road, Reading, RG2 9AX, UK
[3] Met Office, FitzRoy Road, Exeter, Devon, EX1 3PB, UK
[4] Department of Geography, Loughborough University, Loughborough, LE11 3TU, UK
[5] Irish Climate Analysis and Research UnitS, Department of Geography, Maynooth, Ireland
*Correspondence to:* Katie A Smith (k.a.smith@ceh.ac.uk)

**Abstract.** Hydrological models can provide estimates of streamflow pre- and post- observations, which enable greater understanding of past hydrological behaviour, and potential futures. In this paper, a new multi-objective calibration method was derived and tested for 303 catchments in the UK, and the calibrations were used to reconstruct river flows back to 1891, in order to provide a much longer view of past hydrological variability, given the brevity of most UK river flow records which began post-1960. A Latin Hypercube sample of 500,000 parameterisations for the GR4J model for each catchment were evaluated against six evaluation metrics covering all aspects of the flow regime from high, median and low flows. The results of the top ranking model parameterisation (LHS1), and also the top 500 (LHS500), for each catchment were used to provide a deterministic result whilst also accounting for parameter uncertainty. The calibrations are generally good at capturing observed flows, with some exceptions in heavily groundwater dominated catchments, and snowmelt and artificially influenced catchments across the country. Reconstructed flows were appraised over 30 year moving windows, and were shown to provide good simulations of flow in the early parts of the record, in cases where observations were available. To consider the utility of the reconstructions for drought simulation, flow data for the 1975/76 drought event were explored in detail in nine case study catchments. The model's performance in reproducing the drought events was found to vary by catchment, as did the level of uncertainty in the LHS500. The Standardised Streamflow Index (SSI) was used to assess the model simulations' ability to simulate extreme events. The peaks and troughs of the SSI timeseries were well represented despite slight over or underestimations of past drought event magnitudes, while the accumulated deficits of the drought events extracted from the SSI timeseries verified that the model simulations were overall very good at simulating drought events. This paper provides three key contributions: 1) a robust multi-objective model calibration framework for calibrating catchment models for use in both general and extreme hydrology; 2) model calibrations for the 303 UK catchments that could be used in further research, and operational applications such as hydrological forecasting; and 3) ~125 years of spatially and temporally consistent reconstructed flow data derived that will allow comprehensive quantitative assessments of past UK drought events, as well as long term analyses of hydrological variability that have not been previously possible, thus enabling water resource managers to better plan for extreme events, and build more resilient systems for the future.

## 1 Introduction

Hydrological extremes, and associated flood and drought events, threaten security of water supply, food supply, livelihoods and welfare (Kundzewicz and Matczak, 2015). Managing the impacts of both rainfall excess and deficit on the hydrological system poses a significant challenge for authorities and water resource managers across the globe. These challenges are set to become more acute in future: the latest projections for Europe suggest increasing hydrological variability with more severe extremes (Collet et al., 2018; Guerreiro et al., 2018; Teuling, 2018) and further reductions in low flows in many regions (Wilby

and Harris, 2006; Christierson et al., 2012; Prudhomme et al., 2012; Kay et al., 2018; Marx et al., 2018). Increasing demands due to a growing population and socioeconomic changes also imply growing pressures on water resources in the future, necessitating considerable investment in long-term strategic water resources planning and adaptation (Committee on Climate Change, 2017).

Understanding extremes of the past can help us prepare for future extreme events. Drought characteristics of events in the recent past can be used to stress test water supply systems (Mens et al., 2015), a practice that is commonly applied in UK water resource management and drought plans (e.g. Southern Water, 2013 pp. 50-61; Northumbrian Water, 2017 pp. 20-21). Similarly, drought severity estimates of past events have been used to investigate the impacts of increased drought frequency on water supply vulnerability (Herman et al., 2016). There is a growing trend towards testing water supply systems against

events worse than those experienced, using either scenario-based methods (e.g. Stoelzle et al., 2014; Anderton et al., 2015) or stochastic approaches to generate simulated droughts with credible characteristics (e.g. Atkins, 2016). In addition, short-term water management planning can benefit from seasonal forecasting of reservoirs inflows and streamflow volumes (Prudhomme et al., 2017), so that periods of water deficit can be known in advance and appropriate measures put in place to manage resources and mitigate impacts. However, these methods are all dependent on having a good understanding of past variability

and long hydrometric records which are used to train and validate stochastic approaches, and to create tools that enables the simulation of river flows as accurately as possible under a range of varied climate conditions.

Observations of global streamflow are sparse prior to the 1950s, with less than 20% of stations in the Global Runoff Data Centre (GRDC, 2019) beginning pre-1950. Post 1960, the streamflow network expanded rapidly, a pattern that is mimicked by the UK gauging network, where 100 gauging stations in 1950 have increased to over 1300 today. Qualitative data sources

and long rainfall records can identify significant drought events in the pre-instrumentational period (Pfister et al., 2006; Marsh et al., 2007; Brázdil et al., 2016). However, these cannot be used to determine whether these events were more or less severe in hydrological terms than those on the observational record, and there is a need for temporally and spatially coherent flow timeseries to allow systematic assessment of extreme events.

Meteorological records of rainfall and temperature generally extend further back than hydrological data, often providing data

from the turn of the 21st century (New et al., 2000), and occasionally as far back as the mid-20th century. Modelled climate reanalysis data (e.g. Compo et al., 2011), and long term reconstructed climate datasets (e.g. Casty et al., 2007) have been developed for use in scientific research, and can be fed into hydrological models to quantitatively reconstruct river flows beyond the limits of the observational period. In the UK, quantitative reconstructions of river flows using simple hydrological models have previously been conducted, but only for a handful of catchments (e.g. Jones and Lister, 1998; Jones et al., 2006).

Regional flow reconstructions have been used to explore the implications of drought events on water resources (e.g. Spraggs et al., 2015). Drought reconstruction has also been conducted in other countries using proxy data (Jones et al., 1984; Cook et al., 2015), precipitation data (Noone et al., 2017, Ireland), soil moisture models (Wu et al., 2011, China), and hydrological models (Caillouet et al., 2017, France). Generally, however, there are few extant studies that use hydrological models to derive plausible historical sequences.

Catchment hydrological models are tools that can generate streamflow time series from meteorological time series data, to provide continuous proxy river flow data that is otherwise not directly available. They can be used to extend flow records, creating very long sequences that extend back beyond the initiation of the observational network. Such long timeseries can enable thorough analysis of past variability and frequency of severe events (e.g. Caillouet et al., 2017); be used as vital input to short range and seasonal forecasting (Day, 1985; Harrigan et al., 2018), providing valuable early warnings and help

preparedness; or for future projections for long term planning accounting for possible future non-stationarity, for example due to global warming (e.g. Collet et al., 2018).

Calibrating a hydrological model for multiple purposes, e.g. flow reconstruction and forecasting, for high, low and average flows, requires careful consideration. Currently, models are typically calibrated to minimise a specific type of error against observations, measured by an "evaluation metric" also known as an "objective function". Commonly used metrics, such as the

85 Nash Sutcliffe Efficiency (Nash and Sutcliffe, 1970) or Root Mean Squared Error, tend to focus on the correct estimation of high flows (Krause et al., 2005; Dawson et al., 2007), whilst more general metrics, such as Mean Absolute Percent Error and Percent Bias are also used to more systematically optimise the flows and the water balance respectively. There are few examples focusing on optimising low flow simulation. Most commonly, a single objective function is used, implemented using automatic algorithms to find a deterministic parameterisation of the model. Such algorithms are commonly categorised as

"local" (e.g. PEST, Kim et al., 2007) or "global" (e.g. SCE, Duan et al., 1993), some examples of which have been compared by Wallner et al. (2012). However, seeking a single "optimum" parameter set to describe the observations has been argued to be a misconception with theoretical catchment models (Beven, 2012). The need for calibration techniques to maximise hydrological model performance against multiple elements of the flow regime has however been recognised, and multi-objective optimisation methods have been advancing since the turn of the century, though few studies explore more than three

objectives (Efstratiadis and Koutsoyiannis, 2010). Multi-objective optimisation commonly involves seeking Pareto-optimal solutions that find a compromise between objective functions (e.g. Shafii and De Smedt, 2009; Kamali et al., 2013; Jung et al., 2017). Multi-objective methods may also be used to optimise more than one hydrological variable (e.g. Mostafaie et al., 2018). In addition, utilising multiple model parameterisations have been advocated to account for "equifinality" – that many different parameterisations may produce equally adequate simulations of past observations (see, for example: Beven and

Binley (1992); Beven (2006)).

Here, we develop a framework to establish a national network of catchment hydrological models, and evaluate their application to the reconstruction of hydrological time series, with application to the UK over the period 1891 to 2015. The aims of this research are to:

- o Develop a robust method for multi-objective model calibration suitable for use in simulating streamflow with associated
uncertainty.
- o Apply that method to reconstruct historic streamflow time series from the 1890s across the UK,
- o Examine the performance of these time series where observations are available, and
- o Explore the potential for application of these time series in evaluating drought events.

This paper first outlines the datasets in Section 2, before detailing the modelling methods in Section 3. Section 4 provides the

110 results on the performance of the model reconstructions compared with streamflow observations both generally, and during drought events. Section 5 discusses the potential limitations of this work, and suggests directions for further research, before the paper is concluded in Section 6.

## 2 Data

The hydrological model employed in this study (see section 3.1) requires rainfall and potential evapotranspiration data to run,

and observed flow data for calibration and validation. Means of access to the datasets used in this study are described in the Data Availability section at the end of the paper.

## 2.1 Catchment Selection and Flow Data

A diverse set of 303 UK catchments were selected for model calibration. Initially, 395 stations were considered, from the near-natural catchments suitable for low flow analysis from the UK Benchmark Network (Harrigan et al., 2017), and those which
are part of the National Hydrological Monitoring Programme (https://nrfa.ceh.ac.uk/nhmp), which are of particular interest for operational water situation monitoring. Catchments were required to have a minimum of 32 years of observational daily data from the National River Flow Archive (https://nrfa.ceh.ac.uk/), from 1984 to 2015 for model calibration. Some catchments that suffered repeated or prolonged periods of missing data, truncation of flow measurements, step changes, and artificial influences resulting in unrealistic flow patterns were removed from the catchment selection, resulting in 303 catchments. These
125 catchments had records ranging from 32 to 135 years in length, with an average length of 49 years. The average completeness in the gauged daily flows was 99.2% (with a minimum of 90%, and a maximum of 100%). An additional two flow records were included, the naturalised daily flows for the River Thames at Kingston and the River Lee at Feildes Weir, making 305 flow records from 303 catchments. Throughout this paper, the observed calibrations for these two catchments are presented (rather than the naturalised series), for consistency with the other catchments across the UK. While this paper presents summary
results from the whole network, we also selected a set of nine case study catchments to present results in more detail. The nine catchments (shown in Figure 1), were selected from each of the nine hydro-climatic regions defined in (Harrigan et al., 2017) in order to represent the range of hydro-climatology, hydro-geology, and artificial influence across the country, as well as to explore some of the better and some of the poorer model performances among the 303 catchments used in this study.

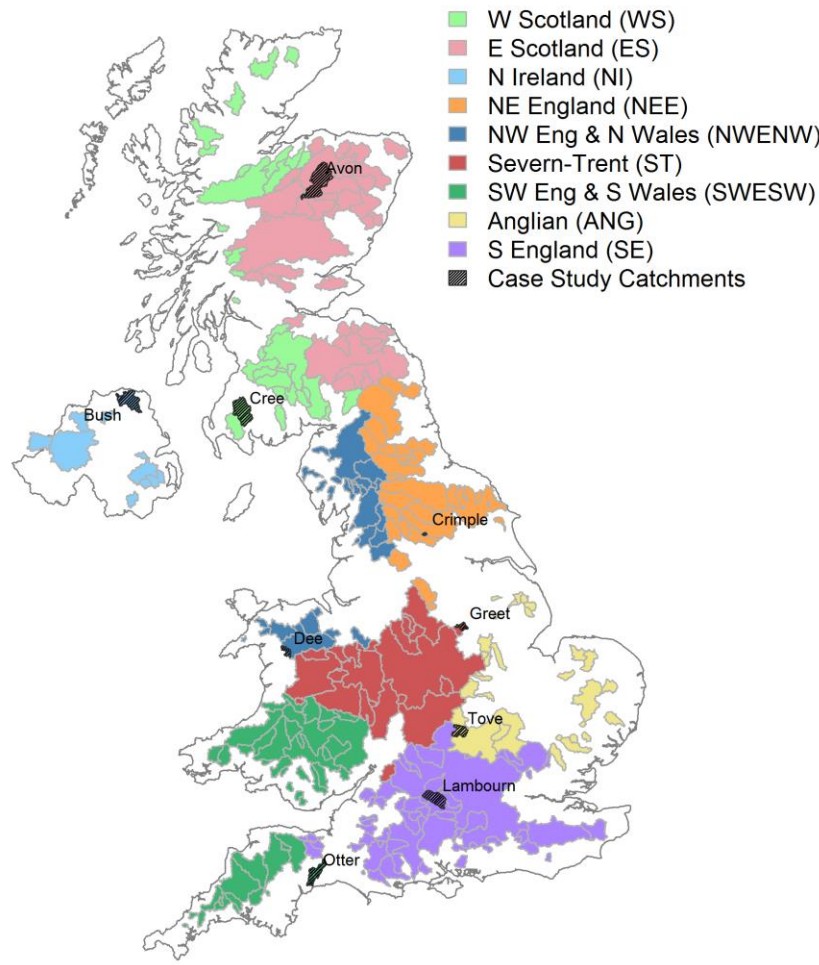

**Figure 1: Map of 303 catchments calibrated. Nine reconstruction case study catchments (one per region) are shown with black hatching.**

## 2.2 Rainfall Data

The daily rainfall dataset used in this study was derived by the UK Met Office as a result of a large data rescue and digitisation programme. The 5km gridded dataset, which covers the period 1891 to 2015, was derived using the same methodology as the UKCP09 data (Met Office, 2017), with interpolation carried out using inverse distance weighting (Perry and Hollis, 2005). The data rescue and digitisation programme added over 200 monthly and 38 daily gauges to the network during the period 1890 to 1910. Catchment averages were derived from the 5km grids, using the catchment boundaries provided from the National River Flow Archive, for use in the hydrological model.

## 2.3 Potential Evapotranspiration Data

As the meteorological variables needed to derive Potential Evapotranspiration (PET) data using the Penman-Monteith equation (Monteith, 1965) are not available prior to 1961, the PET data used for the reconstructions was derived using the McGuinness-Bordne temperature-based PET equation (McGuinness and Bordne, 1972), calibrated for the UK. The temperature data for 1891-2015 were again provided by the UK Met Office following their data rescue programme. A detailed description of the generation of the PET dataset used in this study, following a rigorous analysis of seven temperature based PET equations, four calibration techniques, and seven input temperature data sources/formats, can be found in Tanguy et al. (2018).

## 3 Methods

### 3.1 The GR4J Hydrological Model

The GR4J (Génie Rural à 4 paramètres Journalier) daily lumped rainfall-runoff model (Perrin et al., 2003) was used in this study via the 'airGR' R package version 1.0.2 (Coron et al., 2017). The suite of daily GR models (GR4J, GR5J and GR6J) are being increasingly applied around the world, and GR4J was chosen for several reasons:

1) GR models have been used for streamflow reconstructions previously (Brigode et al., 2016; Caillouet et al., 2017),
2) The GR4J model has demonstrated good performance in a diverse set of catchments in the UK (Harrigan et al., 2018), as well as good performance at simulating temporal transitions between wet and dry periods (Broderick et al., 2016),
3) The GR models are openly accessible, and
4) The model has a low computational demand, and can be run in parallel without manual input requirement.

The model has four free parameters (X1 – X4), requires daily precipitation and potential evapotranspiration data as input, and routes water into two stores: the production store and the routing store. The production store (capacity X1) gains water from effective rainfall, and loses water through evaporation and percolation. Percolated water joins that which has bypassed the production store, and is routed with a fixed split: in which 90 percent is routed via a unit hydrograph (time lag X4), followed by the non-linear routing store (capacity X3); and the remaining 10 percent is routed by a single unit hydrograph (time lag 2*X4). Groundwater or inter-catchment exchange (controlled by X2) is effective on both the routing store, and the flow routed by the single unit hydrograph, and can be positive, negative or zero.

The GR models include an optional snowmelt module, CemaNeige (Valéry et al., 2014). Due to the high computational demand of the snowmelt module, it was decided to calibrate the GR4J model without snowmelt, as only 15 (5%) of the 303 catchments experience a significant fraction of precipitation falling as snow (> 15 %) over the calibration period (Harrigan et al., 2018).

## 3.2 Calibration Strategy

The GR4J model was calibrated for this study incorporating concepts from GLUE type Bayesian approaches (Beven and Freer, 2001), and multi-objective Pareto-optimal solutions (Yapo et al., 1998). The approach consisted of three stages, the details of which are further elaborated in this sub-section: firstly, the feasible parameter space was determined, and sampled using Latin Hypercube Sampling (LHS) (McKay et al., 1979); secondly the model was run, and six evaluation metrics were calculated for each parameter set; and thirdly the top 500 parameter sets for each catchment were selected using a very simple Pareto-optimising ranking approach, accounting for non-acceptable trade-offs (Efstratiadis and Koutsoyiannis, 2010). This method was formalised for several reasons:

1) Latin Hypercube Sampling allowed the systematic sampling of the model parameter space;
2) Multiple evaluation metrics enabled the simultaneous optimisation of several aspects of the flow regime, including general water balance and low flows;
3) Model equifinality (Beven, 2006) could be addressed by accepting multiple "behavioural" parameter sets, and
4) A deterministic "best" parameter set could also be selected.

### 3.2.1 Latin Hypercube Sampling

LHS uses Latin Square theory to ensure that the full range of each parameter is represented regardless of its resultant importance (Cheng and Druzdzel, 2000), whilst maximising efficiency in comparison to simple random sampling approach. An LHS set of 500,000 model parameter sets (parameterisations) for the four model parameters was derived using the MATLAB package 'lhsdesign' (The MathWorks Inc, 2016), using the 'maximin' criterion to maximise the minimum distance between each point. In order to determine what values to ascribe to the upper and lower bounds of the parameters, a smaller experiment using 100,000 model parameterisations was run over 45 catchments as a "first pass". This experiment used parameter limits that could be found in previous literature on the GR4J model (Pushpalatha et al., 2011; Perrin et al., 2003). It was found that good parameter sets for this first pass had storage values (X1 and X3) close to the limits that had been set from the literature. Therefore, in consultation with the developers of the airGR model package, it was decided to widen the ranges of parameter values, and then to increase the number of model parameterisations that were run to account for this increase in the parameter space. The parameter values were sampled from a uniform distribution, using the upper and lower limits given in Table 1. Lower bounds of 0.0001 were ascribed to the two storage parameters to represent a value of 0, without causing division errors.

**Table 1: Sampled Parameter Ranges**

| Model Parameter | Units | Lower Bound | Upper Bound |
|---|---|---|---|
| X1 Production Store Capacity | mm | 0.0001 | 3000 |
| X2 Inter-catchment Exchange Coefficient | mm/day | -20 | 20 |
| X3 Routing Store Capacity | mm | 0.0001 | 2000 |
| X4 Unit Hydrograph Time Constant | days | 0.5 | 15 |

### 3.2.2 Evaluation Metrics

For each of the 500,000 model parameterisations, six evaluation metrics were calculated in order to employ a "multi-objective" approach to cover the full range of the flow duration curve (see Table 2): Nash Sutcliffe Efficiency (NSE), focusses on optimising high flows, Absolute Percent Bias (absPBIAS) maintains the water balance, Mean Absolute Percent Error (MAPE)

and NSE on logarithmic flows (logNSE) measure overall agreement on the full range of flows, and Absolute Percent Error in Q95 (Q95$_{APE}$) and Absolute Percent Error in Mean Annual Minimum on a 30-day accumulation period (MAM30$_{APE}$) focus on low flows. These metrics were calculated over 32 water years 1$^{st}$ October 1982 to 30$^{th}$ September 2014.

Post calibration, the upper and lower daily limits of the 500 top ranking parameterisations (see Section 3.2.3 for details on the ranking process) were used to calculate two further model performance metrics over the full observational record available for each catchment (a maximum of 1891-2014):

- The uncertainty width (UncW) - calculated by taking range of the minimum and maximum LHS500 members each day and dividing it by the midpoint of the LHS500 for that day. The mean of these values was then calculated over the duration of the timeseries, as per:

$$\frac{1}{n}\sum_{i=1}^{n}\left(\frac{ens_{max} - ens_{min}}{ens_{max} + ens_{min}/2}\right)$$

- The containment ratio (ContR) – calculated as the percentage of days that the observations fell within the envelope of the minimum and maximum of the LHS500 ensemble members for that day.

### 3.2.3 Ranking and Selecting Model Parameterisations

In order to optimise six evaluation metrics, the 500,000 model parameterisations were ranked from best to worst by their scores for each metric in turn, and these ranks were then summed to create a total rank. This total, or "basic", rank was used to reorder the parameterisations for each catchment from best to worst, accounting for all metrics. However, the scores of the 500,000 model parameterisations were not normally distributed, and it was found that unacceptable trade-offs between metrics were occurring, whereby nominal increases in one metric were taking preference over quite significant decreases in other metrics. Therefore, a series of thresholds of acceptability were set, as shown in Table 3. A simple iterative search algorithm was then used to re-rank the list according to these thresholds, whilst retaining their original ranks within each threshold group. For example, if the first, third and fourth parameterisations in the basic rank met the hardest threshold for all six metrics, but the second ranked parameterisation did not, the third and fourth would be bumped up the rankings, above the second resulting in a list of [1, 3, 4, 2…]. All parameterisations meeting the hardest thresholds were prioritised before the algorithm switched to search for those in meeting the middle thresholds, and so on. From this final list, the top ranking optimum parameter set was extracted for deterministic model applications, herein referred to as LHS1. Due to the variability of the performance across catchments, where hundreds of thousands of parameter sets met the hardest threshold in some catchments, whilst none met even the softest threshold in other catchments, it was decided that extracting behavioural parameter sets using a 'limit of acceptability' approach after Beven (2006) would not be appropriate. Therefore, a proportion of the sampled model parameterisations, the top 500 (herein referred to as LHS500), were taken forward to provide an indication of parameter uncertainty within the flow simulations. The extent to which the threshold re-ranking influenced the rankings varied by catchment due to the differences in mode performance. Figure 2 shows the NSE and logNSE scores of the 500,000 model parameterisations (though this graph has been limited to show only those with positive scores for both metrics) for the River Greet in Severn Trent Region. This figure demonstrates how the basic ranking system identified 500 parameterisations close to the Pareto front of NSE vs logNSE, however parameterisations with scores that were lower for NSE than logNSE were selected. By applying the thresholds, parameterisations with an NSE lower than 0.4 were rejected, and replaced with others within the acceptable range for all metrics.

**Table 2: Evaluation metrics used for model calibration**

| Evaluation Metric | Equation | Range | Focus |
|---|---|---|---|
| Nash Sutcliffe Efficiency | $NSE = 1 - \dfrac{\sum_{i=1}^{n}(Q_o - Q_s)^2}{\sum_{i=1}^{n}(Q_o - \overline{Q_o})^2}$ | 1 (Perfect) to $-\infty$ | High Flows |
| Absolute Percent Bias | $absPBIAS = \left\| \dfrac{\sum(Q_s - Q_o)}{\sum Q_o} \right\| * 100$ | 0 (optimum) to $\infty$ | Water Balance |
| Mean Absolute Percent Error | $MAPE = \left( \dfrac{1}{n} \sum_{i=1}^{n} \left\| \dfrac{Q_o - Q_s}{Q_o} \right\| \right) * 100$ | 0 (optimum) to $\infty$ | Full Range |
| Nash Sutcliffe Efficiency on log flows | $logNSE = 1 - \dfrac{\sum_{i=1}^{n}(logQ_o - logQ_s)^2}{\sum_{i=1}^{n}(logQ_o - \overline{logQ_o})^2}$ | 1 (Perfect) to $-\infty$ | Full Range |
| Absolute Percent Error in Q95 (flow exceeded 95% of the time) | $Q95_{APE} = \left\| \dfrac{Q95_o - Q95_s}{Q95_o} \right\| * 100$ | 0 (optimum) to $\infty$ | Low Flows |
| Absolute Percent Error in the Mean Annual Minimum on a 30-day accumulation period (30 day moving average) | $MAM30_{APE} = \left\| \dfrac{MAM30_o - MAM30_s}{MAM30_o} \right\| * 100$<br><br>*where* $MAM_{30} = \dfrac{1}{m} \sum_{j=1}^{m} min_j(MovAve_{30})$<br><br>where $MovAve_{30}$ = 30 day moving average<br><br>j = water year in calibration period, m years in length | 0 (optimum) to $\infty$ | Low Flows |

where i is the daily flow value, n is the number of days in calibration period, $Q_o$ is the observed value, $Q_s$ is the simulated values, and $\overline{Q}_o$ is the mean of the observed values.

**Table 3: Thresholds for selecting acceptable model parameterisations**

|  | NSE | absPBIAS | MAPE | logNSE | Q95$_{APE}$ | MAM30$_{APE}$ |
|---|---|---|---|---|---|---|
| Optimum Value | 1 | 0 | 0 | 1 | 0 | 0 |
| Hardest | 0.5 | 10 | 50 | 0.5 | 50 | 50 |
| Middle | 0.4 | 15 | 75 | 0.4 | 75 | 75 |
| Softest | 0.3 | 20 | 100 | 0.3 | 100 | 100 |
| Remainder | <0.3 | >20 | >100 | <0.3 | >100 | >100 |

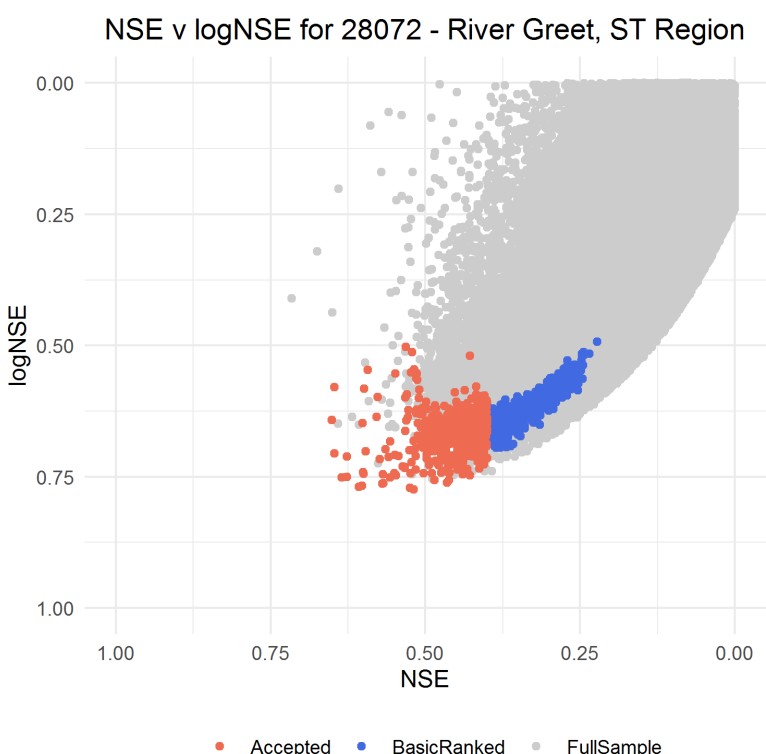

**Figure 2 Nash Sutcliffe Efficiency and log Nash Sutcliffe Efficiency calibration scores for all sampled model parameterisations (grey), the top 500 from the basic ranking process (blue), and the top 500 after the thresholds were applied to negate non-acceptable trade-offs (red). X and Y axes have been reversed, and limited to show only parameterisations that achieved positive scores. Scores of 1 would indicate perfect simulation of the observations; optimal performance is in the bottom left of the graph.**

### 3.3 Flow Reconstructions

Using these 500 model parameterisations per catchment, and the rainfall and potential evapotranspiration data described in Section 2, daily flow reconstructions were produced from January 1891 to November 2015 for the 303 catchments. Details on accessing this data are provided in the Data Availability section at the end of this paper.

### 3.4 Standardised Streamflow Index (SSI)

The application of model results to drought analysis are conducted here using the "Standardised Streamflow Index" (SSI). The SSI has for some years been advocated as an equivalent to the Standardized Precipitation Index (e.g. Vicente-Serrano et al., 2012), being based on the cumulative probability of a given monthly mean streamflow occurring in a given catchment. The procedure involves fitting a statistical distribution to time series of accumulated streamflow over a baseline period, then transforming the data to a normal distribution to produce a dimensionless timeseries of the deviation of flow about the catchment mean. In this study, SSI was calculated using the 12 month accumulation period (SSI-12) and the Tweedie

distribution (Svensson et al., 2017), over the baseline period 1961-2010. A 12 month accumulation period was chosen to provide summaries of long term deficits that were likely to have significant impacts on water resources. The Tweedie distribution, which is a flexible three-parameter distribution that has a lower bound at zero, has been shown to perform effectively for UK river flows, across a wide range of near-natural Benchmark catchments (Svensson et al., 2017).

### 3.5 Drought Accumulated Deficit

Using the Standardised Streamflow Index (SSI), accumulated over a 12 month period, drought events were identified as periods where the SSI was consecutively negative (i.e. below normal) with at least one month reaching an SSI of -1.5 (Barker et al., 2016). The sum of monthly SSIs during these events was calculated to derive the accumulated deficit (e.g. Noone et al., 2017; Barker et al., 2019).

## 4 Results of Model Calibrations

### 4.1 Model Calibration Statistics

The map in Figure 3 shows the threshold (as set out in Table 3) met by the LHS1 runs and the percentage of the LHS500 members that met that threshold. The map shows that the LHS1 runs for 272 of the 303 catchments met the hardest threshold set (shown as triangles). However, there is a lot of variability within these catchments, with 82 demonstrating all of the LHS500 met the hardest threshold (black triangles), whilst 108 have less than 10% of the LHS500 above the hardest threshold (yellow triangles). The LHS1 run for 20 of the catchments met the "middle" threshold, and very few catchments performed worse than this, having <0.4 for NSE and logNSE, >75% for MAPE, $MAM30_{APE}$ and $Q95_{APE}$, and >15% for absPBIAS (5 catchments in the "softest" threshold, shown as circles, and 6 catchments that failed to meet even the "softest" threshold, shown as crossed circles). These localised examples of poor model performance (shown as circles and crossed circles) may be due to the lack of snowmelt processes in the model (in Scotland and North East England), human influences such as abstractions and water transfers or significant groundwater interactions (in Anglian and Southern England). For the Warleggan in Cornwall, poor performance is due to underestimation of peak flows, which may be attributed to an issue in simulating the localised geological outcrops.

Figure 4 shows the results of the six evaluation metrics for each of the 305 flow reconstructions over the calibration period (1982-2014), for both the LHS1 runs and the range of the LHS500. These polar plots confirm the findings of Figure 3, showing that the model performance is generally very good, with most of the LHS1 runs for the 305 catchments satisfying the thresholds defined in Table 3 with ease. This plot allows the assessment of each performance metric individually, and shows that performance varies both between metrics, and across catchments. The poorest scores, where the LHS1 did not meet the softest threshold can be mostly attributed to NSE, but MAPE and $MAM30_{APE}$ each also account for one failed catchment. $MAM30_{APE}$ shows the fewest LHS1 scores below the hardest threshold, and NSE the most. LHS1 points are mostly on the extreme periphery of the absPBIAS and $MAM30_{APE}$ plots, demonstrating very good results, but several catchments deviate quite substantially from this. $Q95_{APE}$ exhibits a similar, but not so strong pattern; whilst the LHS1 points for NSE, logNSE and MAPE are far more scattered. The ranges of the LHS500 scores are also varied, with some very narrow ranges, particularly in the SE region across all metrics. These narrow ranges show that the 500 ensemble members are very similar in performance. Beyond the SE region, the ranges of model performance among the LHS500 do not appear to show any regional pattern, but are generally narrower for the NSE, logNSE and MAPE metrics than absPBIAS, $MAM30_{APE}$ and $Q95_{APE}$. These results show that using this multi-objective calibration procedure, all six of the evaluation metrics were well optimised for the majority of

catchments, providing confidence in the application of the flows derived from these model calibrations across the range of flow values.

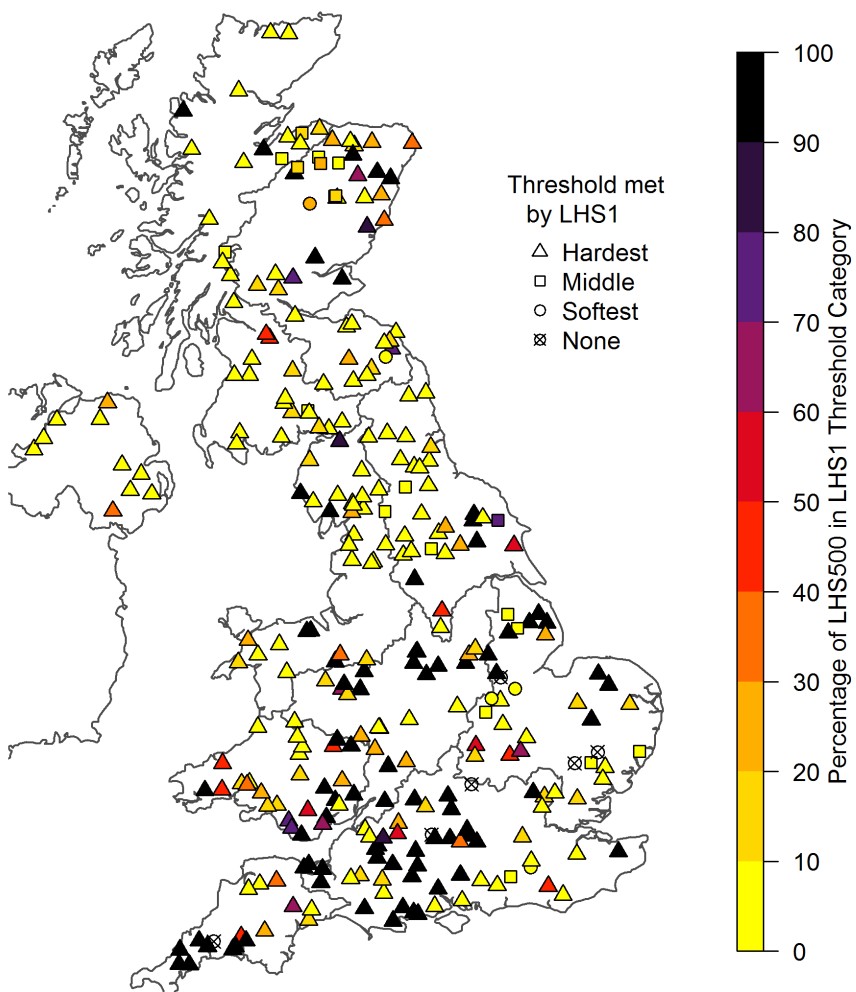

 **Figure 3: The threshold met by the LHS1 model parameterisation (shape), and the percentage of the LHS500 that met that threshold (colour), for the 303 study catchments. See Table 3 for the definition of the thresholds.**

Figure 5 shows the scores of the uncertainty width (UncW) and the containment ratio (ContR) for each of the 303 catchments. The lower the UncW (a narrow range of model results among the 500 ensemble members), and the higher the ContR (a high proportion of the observations fitting within the band of model runs), the more accurate and reliable the simulation is. In these results, there appears to be a correlation between UncW and ContR (Pearson correlation 0.52, with significance, p value 2.2e-16): where UncW is high (which can be seen as poor), the ContR is also high (seen as good), and vice versa. This highlights the need to consider both of these elements when assessing the confidence in the model, as a low UncW with a low ContR would suggest a biased, and under-sensitive model. Catchments with the smallest UncW associated with low ContR are located in central southern England, parts of north-east England, and eastern Scotland. Whilst attribution of the cause of this modelling deficiency is difficult and out of scope here, it is possibly linked with the "flashiness" of the catchment, which can be due to groundwater and human influences (southern England and parts of north-east England), and snowmelt (eastern Scotland). In the majority of the catchments (250 of 303), the ContR is greater than 80%, but the UncW is also greater than the mean flow in 189 of those catchments.

These graphs represent an overview of the performance of the model calibrations across the UK. The model performance for individual catchments, as well as timeseries of the reconstructed flow data from 1891-2015, can be explored in more detail using the interactive web application at https://shiny-apps.ceh.ac.uk/reconstruction_explorer/.

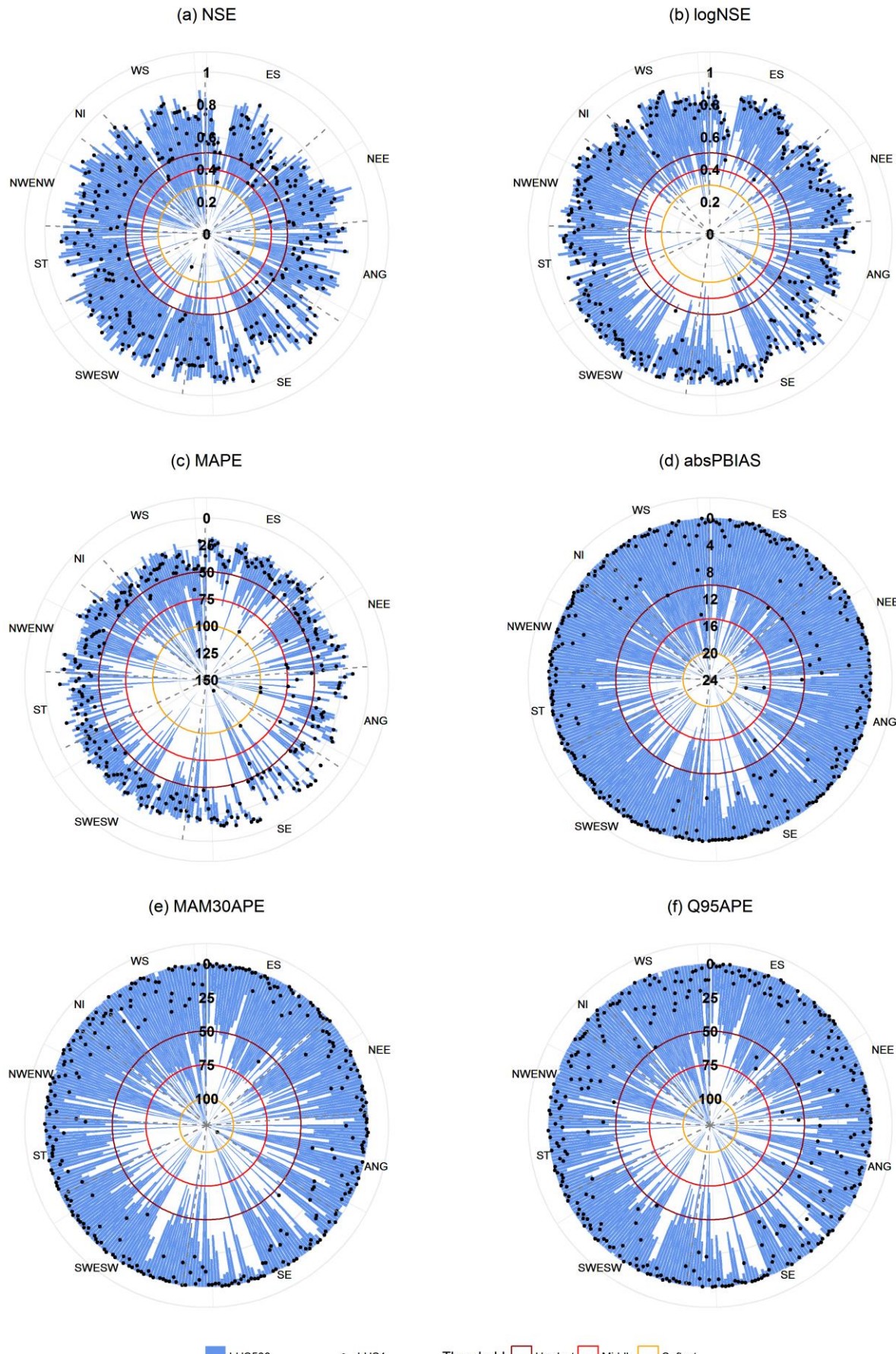

**Figure 4: Polar plots of the scores for six evaluation metrics over the calibration period 1982-2014. Each blue bar and associated dot represents one of the 303 catchments, plotted around the perimeter of the circle, grouped by hydrometric region: see Figure 1 for region abbreviations. Dark blue dots represent the LHS1 run, and blue shaded bars represent the range of the LHS500. The score is shown on the radial axis, with the outside of the circle representing best model performance.**

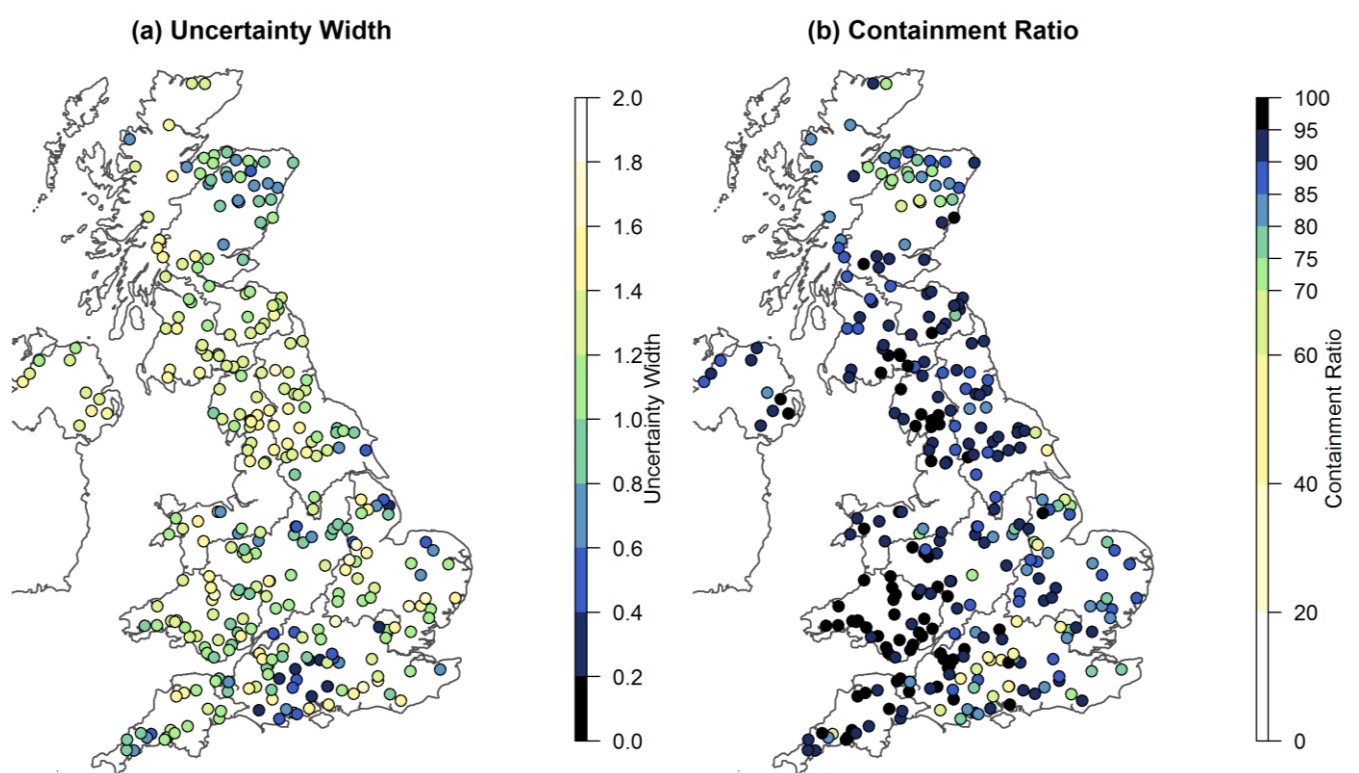

**Figure 5 (a) Uncertainty width, and (b) Containment ratio over the calibration period (water years 1982-2014) for all 303 study catchments. In these maps, darker blue colours represent better scores.**

## 4.2 Thirty Year Model Validation Statistics

In order to evaluate the integrity of the reconstructed flow series, in the earlier pre-calibration parts of the record, the six evaluation metrics for the LHS1 runs specified in Table 2, as well as the uncertainty width and the containment ratio for the LHS500, were calculated over thirty year moving windows for all water years where flow observations were available. These results have been plotted as polar heatmaps in a similar way to the polar plots showing the evaluation metrics over the calibration period. Figure 6 shows the heatmap for $Q95_{APE}$, whilst all eight heatmaps are provided in Supplementary Figure S1. In these figures, the catchments with longer observational timeseries are shown as longer bars that originate nearer the centre of the circles. Here it can be seen that observations for most catchments began after the 1960s, and only 12 catchments have observations prior to the 1940s. The two longest series in the south-east (SE) region are the Lee at Feildes Weir (plotted at the boundary with Anglian region) and the Thames at Kingston (plotted three catchments further clockwise). Long records can also be seen in the Dee in east Scotland (ES), and the Severn in Severn Trent (ST) region.

In general, across all metrics and catchments, the scores are very stable: where bars are dark or pale, showing good and poorer model performance respectively, they remain similar colours throughout their length. There are some exceptions, which are most notable in the catchments with longer observational records. The Avon at Evesham in ST region, the Dee at Manley Hall in North West England North Wales (NWENW) region, and the Bedford Ouse catchment in Anglian (ANG) region, show reduced model performance earlier in the record, with the bars moving through orange and yellow shades as they stretch towards the centre of the circle. It is worth noting though, that these catchments are not part of the near-natural Benchmark Network (Harrigan et al., 2017), and have had reported issues with inhomogeneity in their observed records as a result of human influences. The Lee at Feildes Weir in SE region (plotted at the boundary with ANG region) also shows variation in performance across most metrics, although in this catchment, the performance is good (plotted in black) at the start and end of the record, with poorer performance (shown in yellow) around the start years of 1920-1940 (evaluation years of 1920 to 1970).

In contrast to this, the Dee at Woodend in East Scotland, and the Severn at Bewdley in Severn Trent region, which have the longest records in their regions, show more temporal stability in the model performances (with black colouring for the whole bar). This, coupled with the generally very stable results over the 20-30 years prior to the calibration period among with the catchments with shorter records, demonstrates that the flow series produced for this study are suitable for use in longer temporal studies, outside of the period of calibration (1982-2014).

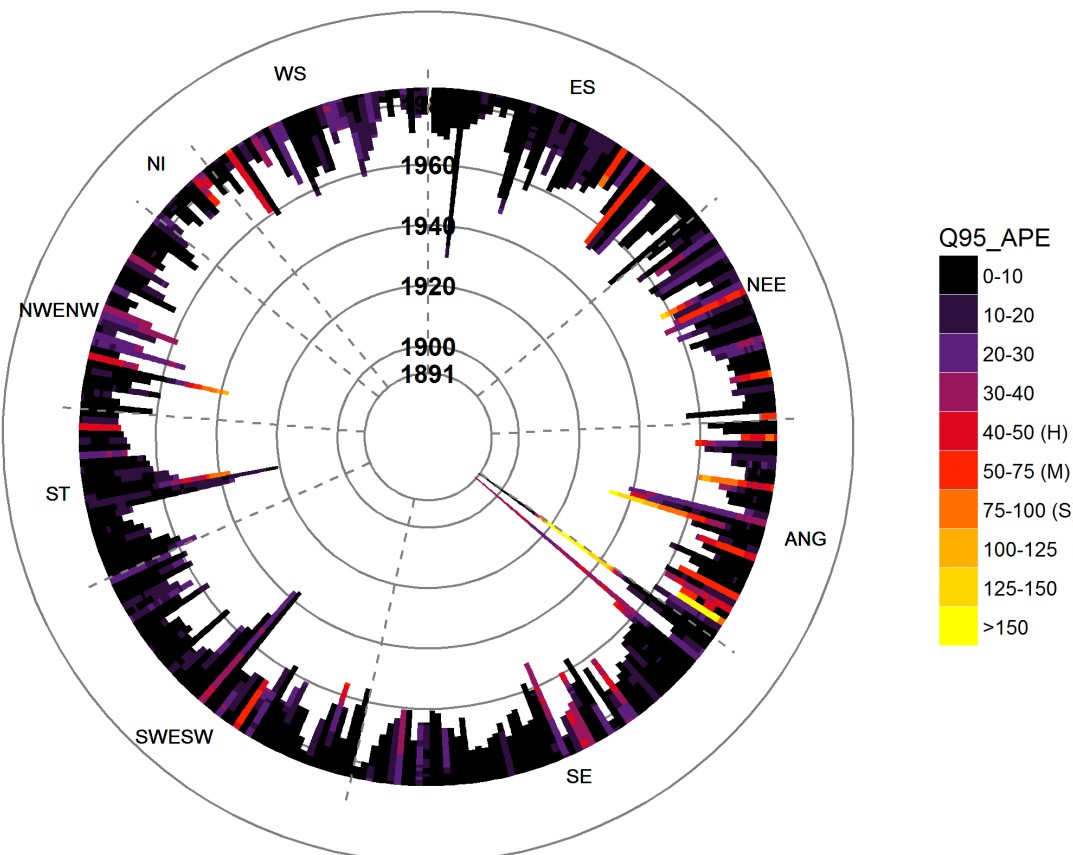

**Figure 6: Polar heatmap showing Q95$_{APE}$ scores calculated over 30 year moving windows for all available water years of observed flow data. Each bar represents one of the 303 catchments, plotted around the perimeter of the circle, and grouped by hydrometric region: see Figure 1 for region abbreviations. The starting year of the 30 year window is represented on the radial axis with 1891 plotted towards the centre of the circle. Catchments with longer observational records have longer bars. The shading of the bars represent the Q95$_{APE}$ scores, with darker colours being optimum. The hardest (H), middle (M) and softest (S) thresholds are labelled on the legend.**

## 5    Reconstructions of Drought Events

In this section, the nine case study catchments (shown in Figure 1) are used to examine the performance of both the LHS1 and the LHS500 modelled flows at simulating drought events.

### 5.1    The 1975/76 Drought event

The 1975/76 event was chosen as a case study period to test the model's capability to reconstruct drought events. This event occurs before the model calibration period of 1982-2014, and was one of the most severe and widespread droughts of the 20[th] century in the UK (Marsh et al., 2007). Summary statistics showing the model performance for these catchments both during the calibration period (1982-2014), and the ten year period surrounding this significant drought event (1971-1980) are provided

in the Supplementary Information Table S1. It is worth noting that the observational records in the Bush and Crimple did not begin until 1972, nor the Greet until 1974.

### 5.1.1 Flow Timeseries

The plots in Figure 7 show observed and simulated monthly flow for the years 1971 to 1980. Here, the simulations in each catchment capture the variability of the observational record well, however the model results show differing ensemble ranges between catchments. The range of the LHS500 members (referred to as the uncertainty width in Table S1) appears in the graphs to be much wider in the Avon, Greet, and Tove than in the Dee, Cree and Lambourn, but this is not reflected in the statistics. This is likely due to the higher inherent variability or "flashiness" in the Dee and Cree over the Avon that is affecting the visualisation of the uncertainty width (UncW) in the graphs. The Lambourn does have a particularly narrow UncW (0.23 over the ten year period), but the Dee and the Cree have some of the largest UncW (1.44 and 1.46 respectively), with the Crimple showing the highest (1.52). It is evident that where the UncW is low, the observations are more likely to fall outside of this range; with the exception of the Lambourn at 52%, the ContR across the catchments for this period is very high (exceeding 73%), and there are very few instances where the observations fall outside of the range of the model ensemble members.

In the Crimple, the UncW is especially wide during low flow events, and the observations lie very close to the lowest of these model runs; however the LHS1 run lies close to the observational flow values. In other catchments, such as the Otter, the observed and LHS1 flows sit more centrally within the range of the LHS500. In the Avon, the observations sit centrally within the uncertainty range, however the LHS1 run overestimates low flows. The LHS1 flows for the Cree tend to underestimate the low flows. The Avon and the Bush display poor scores in the low flows metrics $MAM30_{APE}$ and $Q95_{APE}$ compared with other catchments during the 1971-1980 period. The inclusion of low flows evaluation metrics in the LHS calibration procedure does not appear to have heavily impacted the performance of the model during high flows. The high flows that followed the 1975/76 drought event are very well simulated, with the exception of the Lambourn and the Greet where there are slight discrepancies in the monthly peak flows.

Daily flows for Jan 1975 – Dec 1976 (shown in Figure 8) highlight the difference in variability between the catchments in the northern and southern parts of the UK. The variability is generally well simulated, though the GR4J model exhibits some difficulty in simulating the low flow variability in the southern catchments, with very little inter-monthly variability in the simulated discharge, although significant peaks are identified among the ensemble members. Note that the abnormal peaks of the observational record on the Lambourn in Sep-Dec of both 1975 and 1976 are the result of the West Berkshire Groundwater Scheme (WBGS) that was implemented during the drought to alleviate the extreme low flows, and are not accounted for in the model which has no human influence representation. Generally, LHS1 simulations are low among the LHS500 runs in the Cree, Bush, the Crimple (as seen in the monthly plots), but are close to the observations. This indicates that selecting the "best" simulation where a deterministic result is needed is more appropriate, in these cases, than extracting a mean or median from the ensemble.

As with the monthly flows, the Avon and the Bush show systematic overestimation of the low flows by the LHS1 run, whilst the Cree shows underestimation of low flows, with the exception of the most extreme low flows in Jul-Sep 1976. These mixed results that can be seen for the nine case study catchments highlight the variation in model performance among the 303 modelled catchments, and emphasise the need for users to carefully appraise the evaluation metrics of the flow simulations for the catchments they are investigating. However, these catchments were deliberately selected to explore these variations, and the results shown in Section 4 demonstrate that the model performs well across a wide range of different catchment types at the national scale.

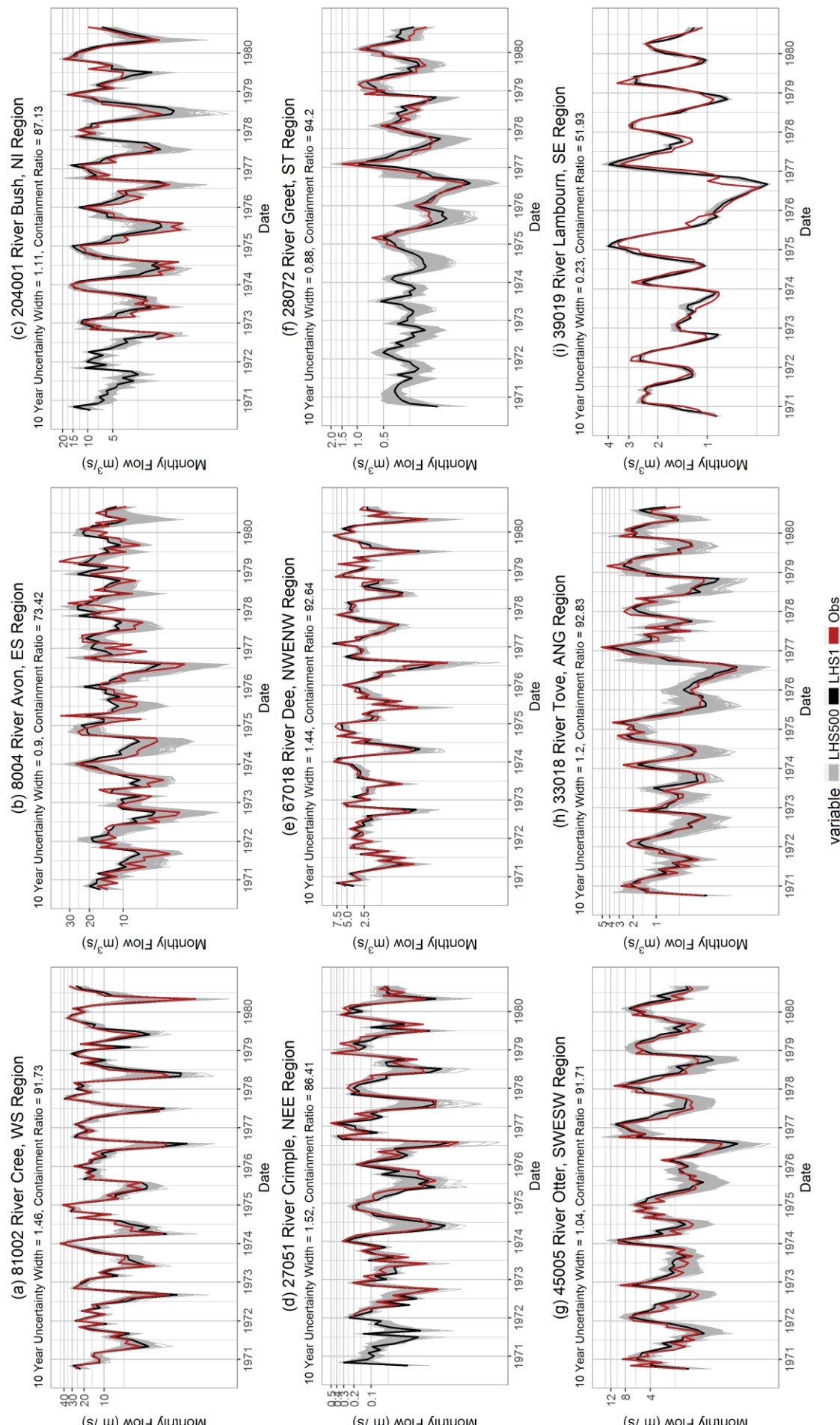

**Figure 7: Monthly mean flows for 1970-1980. Each of the LHS500 ensemble members are shown in grey, with the LHS1 run shown in black, and the observations shown in red. The y-axis is presented on a log scale in order to allow better visualisation of low flows.**

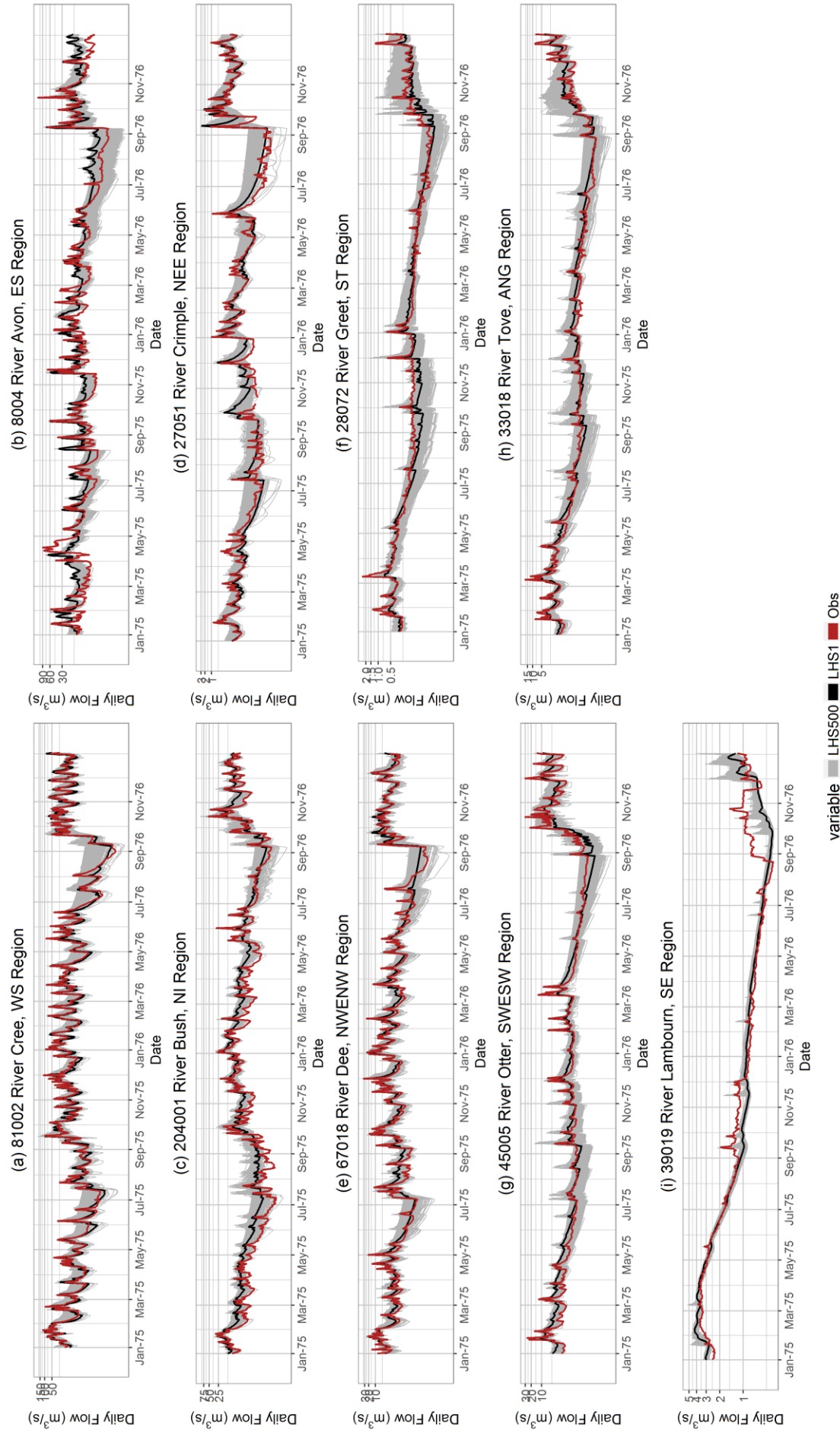

**Figure 8: Daily flow timeseries for the 1975-1976 drought event in each of the nine case study catchments. The y-axis is presented on a log scale in order to allow better visualisation of low flows.**

### 5.1.2 Standardised Streamflow Index (SSI)

SSI data for the LHS1 runs have been calculated for all 303 catchments, and are freely available (Barker et al. (2018), see Data Availability), but have also been calculated here for the LHS500 for the nine case study catchments. These data are used to evaluate how well the ensemble simulations reproduce the drought event accumulated deficit. For low flows, we consider SSI values of -1 to -1.5 to indicate a moderate hydrological drought, -1.5 to -2 a severe drought, and SSI values below -2 an extreme drought (after Barker et al., 2016; McKee et al., 1993).

Here, the SSI timeseries for the same ten year period (1971 to 1980) are appraised, and shown in Figure 9. The uncertainty widths (UncW) in the SSI plots shown vary substantially between catchments and directly reflect the ranges seen in the flow timeseries: with the Lambourn showing a very low UncW from the LHS500, whilst the Greet, Tove and Otter show a wider range. In the Lambourn, Dee, and Bush catchments, the SSI derived from the observations frequently fall outside of the range of the LHS500, showing a low containment ratio (ContR). This behaviour is more pronounced in the SSI timeseries than the flow timeseries. The Dee catchment, for example, produced a ContR of 92.6% for the daily flow data over 1971-1980, but the SSI-12 ContR is just 30%. It is noticeable that the UncW of the SSI data are fairly even throughout the timeseries, whilst in the flow data, they appear to be wider during the more extreme high and low flow periods. There are two factors which may have contributed to these differences: firstly that the smoothed nature of the SSI-12 reduces the short term variability of the data (the ContR of the SSI-1 are closer to those of the flow data); and secondly, when deriving the SSI, the tails of the fitted distribution are more uncertain than for the average flows, which may result in convergence of the SSI values for the more extreme members of the LHS500 during periods of high and low flows.

For the Lambourn, the negative SSI values (below normal flows) are underestimated and the positive SSI values (above normal flows) are overestimated showing the model is overemphasising the extreme events. In the Avon catchment the most extreme SSI deficit occurs in 1973, and the 1976 event is classed as "severe", but not "extreme" for the observations and all but a few of the LHS500. The deficit in 1973 is simulated as being more extreme than the observations but the 1976 event is better captured. The uncertainty range in the Greet catchment is very wide, particularly for the SSI peak (drought termination) in 1977, however the 1976 SSI deficit has a lower range among the LHS500. For the Tove, the SSI of the 1976 drought event is well simulated, as are those for the Crimple, despite some underestimation of SSI at other times in the 1970s. The Otter shows very good simulation of SSI-12 during this ten year period.

SSI timeseries plots over the longer period 1975 to 2015 are provided in Figure S2 of the Supplementary Information. These plots show that although the exact magnitudes of the SSI deficits and excesses are not always captured by the model in some of the poorer performing catchments, the pattern of the SSI-12, the shape of the peaks and the troughs are very well represented.

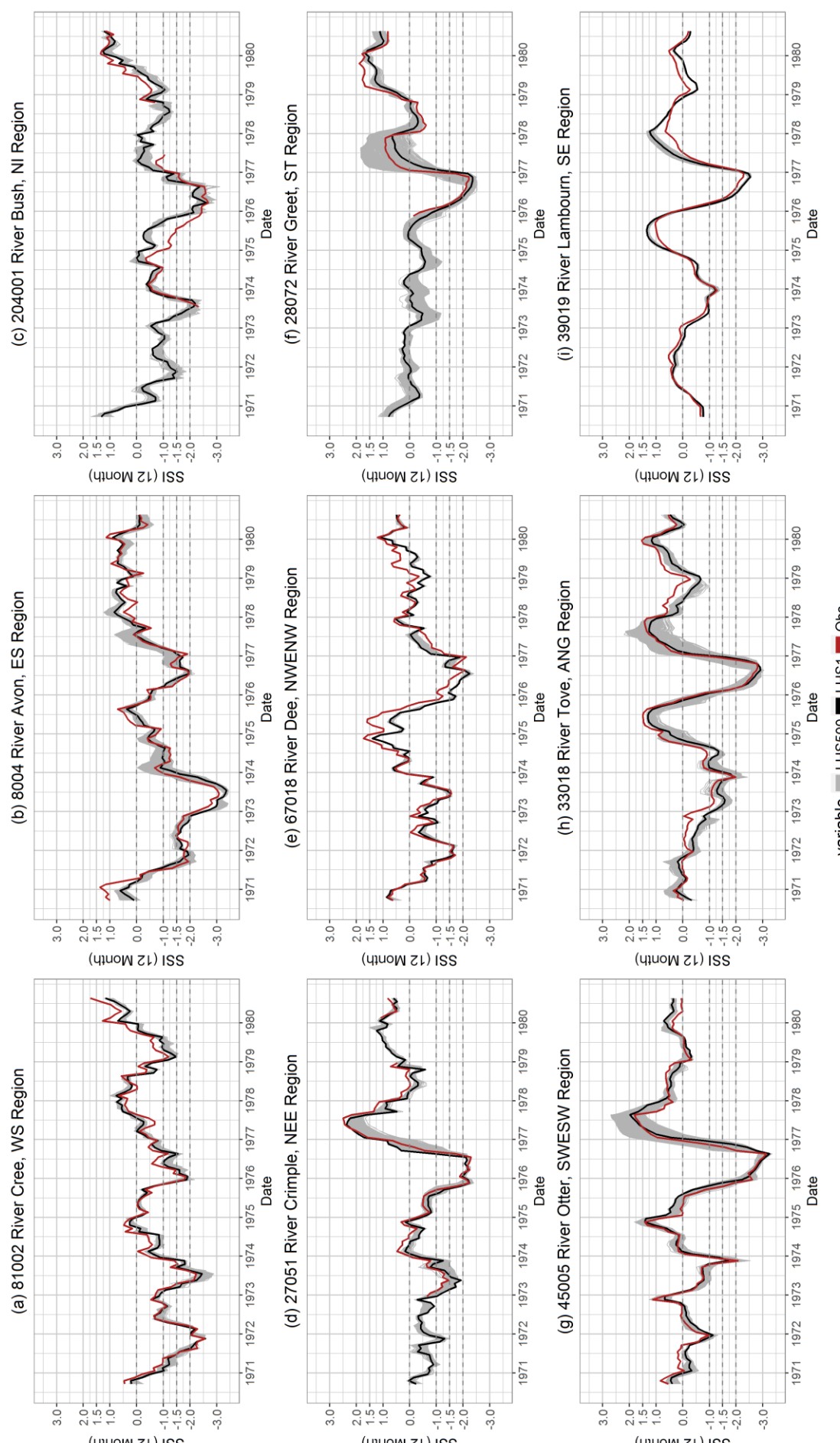

**Figure 9: Standardised Streamflow Index (SSI) using a 12 month accumulation period for nine case study catchments over the years 1971-1980. SSI values of 0, as well as those representing moderate (-1), severe (-1.5) and extreme (-2) drought are shown as dashed horizontal lines on the y-axis.**

## 5.2 Drought Event Accumulated Deficits

This section explores the accumulated deficits of extracted drought events between 1975 and 2015 (the common observed period for all nine catchments), which are presented in Figure 10.

This plot shows that drought events are generally in good synchrony across the country. For these nine catchments, four major nationwide drought events using SSI-12 are evident: 1975-1978, 1989-1993, 1997-1998, and 2004-2006. Regional droughts include 1984 in the northern catchments, and 2010-2012 which affected England and Wales, but not Scotland and Northern Ireland. There appears to be a relatively "drought poor" period in the south between 1977 and 1988, whilst the north shows a lack of droughts in the more recent period of 2006-2015.

The observed events are very well captured by the model simulations. There are only four out of a total of 40 observed drought events across all nine catchments that are not detected by the simulated drought events: an event in 1992 in the Crimple, 1994 and 2004 in the Dee, and 2006 in the Cree. In each of these cases, the SSI of the model simulations fall below -1, but do not reach -1.5 (see Figure S2), suggesting an overestimation of low flows, and therefore a slight underestimation of the drought deficit for this event. In contrast, there are some drought events that are identified from the model simulations that are not evident in the observed record, for example 1998 in the Avon and the Bush. In these events the model underestimates the flow, and therefore overestimates the drought deficit. In the Bush, this underestimation of flow continues during the low flow periods of 2002 and 2003-2006.

In terms of timing and deficit, the 1995-1998 drought event demonstrates the most confidence among the simulations. The Crimple catchment shows some uncertainty about the timing of each of the events, and the majority of the LHS500 model simulations place the 2004-2006 event later than the observation. In Figure S2, it can be see that this is due to the fact that the intensity of the 2005 deficit was overemphasised by the model. Similarly, in Figure 10, the 1975-1978 event in the Bush shows a wide range of mid-point dates (centre of the circles), and the deficit also varies. Overall, the deficits of the events are well captured by the modelled data: for example, the 2004-2006 event in the north showed smaller deficits than the 1975-1978 event, and the modelled deficits reflect these differences. The modelled results for the 1997-1998 event in the Greet show two possible event timings, and the thickness of the circles indicate some differences in the simulated accumulated deficit among the model parameterisations, though these differences are relatively small.

On balance, the pattern of drought events is well simulated by the GR4J model, despite some small differences in magnitude and timing, with magnitude being better estimated than the timing. These results demonstrate that, despite the issues seen in the SSI timeseries plots, the dataset can provide good estimates of drought events and their characteristics. This highlights the potential of the model to reproduce hydrological drought events using just precipitation and evapotranspiration data, and shows that the reconstructed flow timeseries will be valuable in appraising historic hydrological droughts over a longer period and wider spatial domain than the observations that are available.

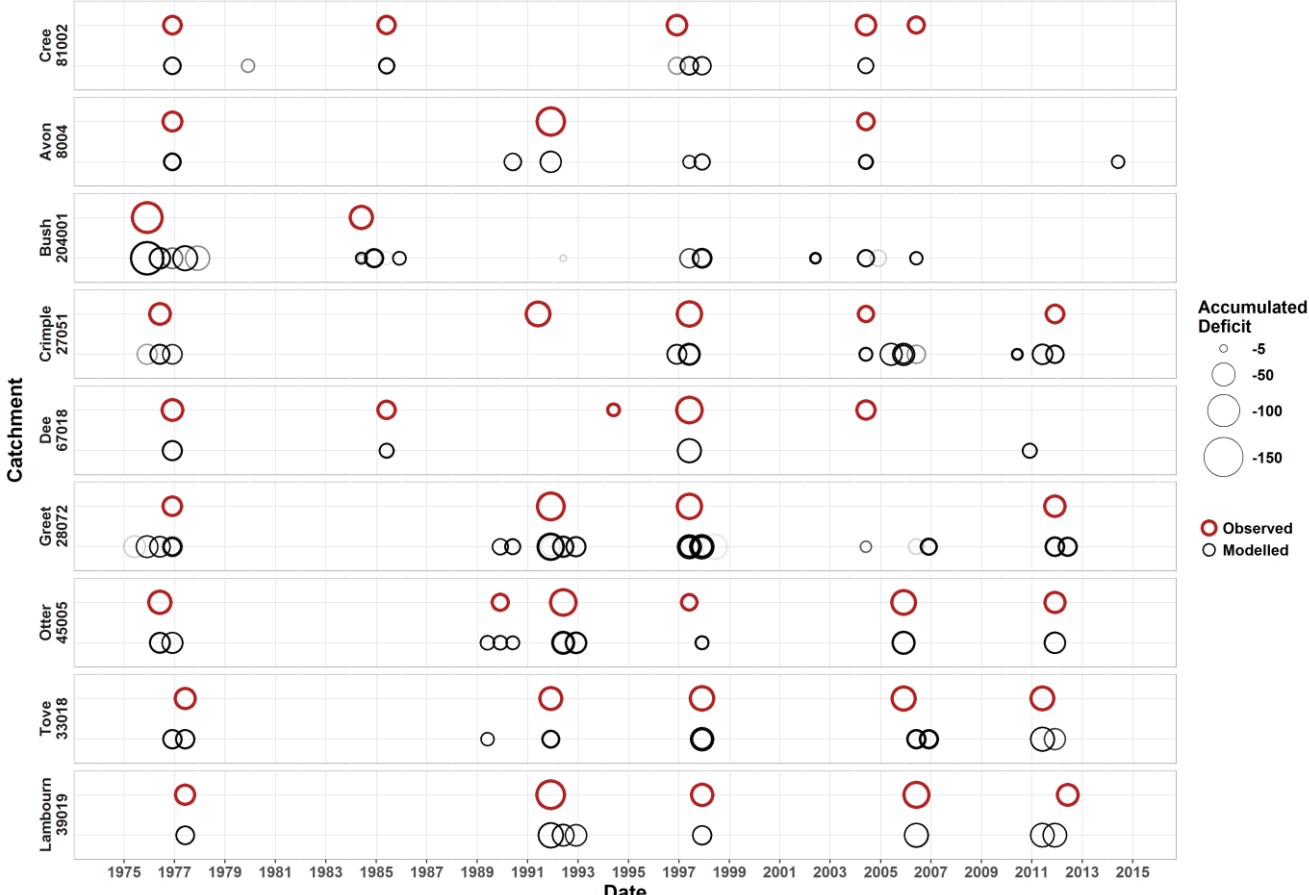

**Figure 10: Accumulated deficits of extracted drought events (using a threshold of SSI < -1.5) for nine case study catchments over the years 1975 to 2015. Circles are plotted along the x-axis according to the date of the mid-point of the extracted drought event. The circle size represents the magnitude of the accumulated deficit. Drought events extracted from the observed data are shown in red (with a thick circle for visibility). Events extracted from the 500 ensemble members are shown with thinner black circles (these circles are semi-transparent, where these circles appear black, multiple ensemble members are simulating the event, and where they are thick, the ensemble members show different accumulated deficit values). Multiple, overlapping black circles suggest discrepancies in the timing of the drought event among the ensemble members.**

## 6   Discussion

The multi-objective calibration framework presented in this paper has produced modelled flow data with demonstrable high performance across a wide range of available observed records. This framework has been developed to enable nationally and temporally coherent flow simulation that can be applied to a wealth of applications, past, present and future. In this paper, the calibration framework has been applied to a wide range of catchments across the UK, allowing for a detailed exploration of model performance across different hydrological regimes. Two potentially limiting factors in model performance were highlighted in this study: snowmelt, and human influences.

The airGR snowmelt module was not employed in this study as only 15 of the 303 catchments showed snowmelt fractions greater than 0.15 (15%). These catchments were located along ten rivers, all in Scotland. Despite the lack of snowmelt processes here, all of the catchments met at least the "softest" evaluation thresholds set out in Section 3.2.3, with six, eight and one catchment meeting the hardest, middle and softest thresholds respectively. This implies that snowmelt only causes modelling issues for high altitude Scottish catchments.

Human interactions are a common problem in hydrological modelling that remain largely understudied (Calvin and Bond-Lamberty, 2018). Whilst global scale models have been advancing in socio-hydrology, making use of satellite information and governmental estimates of total water consumption, the data to support such endeavours is lacking (Bierkens, 2015). Small

scale catchment models would need to rely on significant amounts of abstraction and licencing data as well as reservoir operation procedures, the details of which are often sensitive and/or unavailable. The lack of abstraction processes in GR4J is likely to be responsible for some reduced model performance, particularly in the regions of Anglian and Southern England. The loss function (parameter X2 "inter-catchment exchange coefficient") of the GR4J model can account for some systematic losses or gains, however human influence is often non-stationary (e.g. construction and operation of reservoirs, irrigation and water transfer schemes). For the Lee at Fieldes Weir and the Thames at Kingston naturalised river flow data, which attempt to remove the impact of human activity on the observed flow, were available. Whilst not included in this paper for consistency with the other 301 catchments, calibration scores were slightly better for the naturalised flow data in these catchments, though both naturalised and observed calibrations easily met the hardest thresholds. An alternative approach is to focus studies on the "near-natural" catchments, which are deemed to have minimal human influence. Of the 303 catchments included in this study, 115 are classified as near-natural and are part of the Low Flows Benchmark Network (Harrigan et al., 2017). Since many of the UK's most significant catchments are heavily influenced, they were not excluded from this study, and the model does successfully manage to implicitly account for human influences in these large rivers. Localised issues in the model's performance, and therefore the quality of the reconstructed flow data, highlights the need for users to take caution when choosing a catchment from this set of 303. Depending on their needs, an alternative nearby catchment where model performance is better, may be more suitable if model performance is poor in the initially selected catchment.

The modelling framework developed in this study has explored model parameter uncertainty in order to account for equifinality (Beven, 2006). 500,000 parameter realisations were run, and the best 500 of these were selected for each catchment to allow for uncertainty quantification in applications of these flow data. Here, the uncertainty in the model runs was shown to vary more between catchments than over time (from 1890-2015, where long observational records were available). Whilst model parameter uncertainty was considered in this study, further sources of uncertainty can contribute to variations in model performance, including: model input data (precipitation and PET), flow data used for model calibration, and the choice of hydrological model (Smith et al., 2018b).

The impact of precipitation uncertainty has been shown to be more significant than PET in hydrological modelling (Paturel et al., 1995; Bastola et al., 2011; Guo et al., 2017). Perry and Hollis (2005) and Legg (2015) state that the accuracy of gridded data is dependent on the density of the rain gauge network, with greater errors associated with sparse coverage. Therefore errors in the reconstructed precipitation data applied in this study will be higher in the early part of the record when the station density was lower. Since the model is calibrated to the more recent period 1982-2014, uncertainty from the rainfall data may propagate through to the flow reconstructions in the early part of the record. However, from the moving window analysis of model performance (see Figure 6), there does not appear to be significant degradation in the quality of flow simulations in the early part of the record. Tanguy et al. (2018) considered the impact of poorer quality and lower density of temperature data on the derivation of the PET dataset that was employed in this study and concluded that, whilst PET is an important variable for predicting runoff, the influence of degraded PET input that result from low quality temperature data on runoff simulation can be limited by the adequate calibration of hydrological models (Bai et al., 2016; Seiller and Anctil, 2016). Thus, the Tanguy et al. (2018) PET dataset is considered suitable for use in hydrological models, especially if they are calibrated to this dataset.

Uncertainties may also arise from the observational flow data used to calibrate models. Uncertainties from the precision of the instruments that measure the water level (stage), and uncertainties from the derivation of the stage-discharge relationship are both particularly sensitive in the extreme flow ranges. For example, a 10mm error in stage measurement at the Q95 flow can result in a 20% error in flow for around a third of the UKs gauging stations (National River Flow Archive, 2018). The dataset used in this study was taken from a reputable source (the NRFA) who in order to minimise such errors, conduct rigorous quality control procedures using both automatic and manual validation procedures annually (Dixon et al., 2013). Nevertheless,

hydrometric data quality does vary across the network and errors tend to cluster in the extreme flow ranges, so hydrometric uncertainty could be influential in some periods in catchments used herein – we recommend users consult the NRFA's extensive station and catchment metadata (available at https://nrfa.ceh.ac.uk/) in conjunction with model performance information (Smith et al., 2018a) when using the flow reconstructions.

Whilst the parameter uncertainty in the model was evaluated here, applying different model types and model structures can also yield dramatically different results. Many multi-model experiments have been conducted to assess the differences between hydrological models (e.g. Warszawski et al., 2014;Vansteenkiste et al., 2014). Similarly, different structures of the same model (e.g. GR4J, GR5J and GR6J) can influence the results. However, Smith (2016) found that model parameter uncertainty can be as wide as that from using different hydrological models, and initial testing of the GR5J and GR6J models showed significant parameter interactions that resulted in poor simulations in many UK catchments. It was therefore decided that considering the parameter uncertainty of the GR4J model would be sufficient to devise an ensemble of flow reconstructions for this dataset and study. Future work will investigate these simulations against a wider set of model runs using other model structures as part of a follow-up study.

The modelling framework developed here has been shown to be fit-for-purpose for drought reconstruction, across a very wide range of catchment behaviours. The reconstructed series can be used to shed light on historical drought occurrence, characteristics (severity, duration, termination, seasonality) and variability. A first exploration of hydrological drought using the reconstructions is presented in a companion paper by Barker et al. (2019). The data can also be used to support drought and water resources planning activities, whether directly or to provide context for stochastic approaches to drought generation. Ensembles of historical drought events can be used to provide insight into the probabilities of the termination of a current event over a certain time period (e.g. Parry et al., 2018). Knowledge of historic events can also be used to explore statistical correlations with atmospheric drivers of droughts that may help predict the onset of events (e.g. Lavers et al., 2015). In these approaches, extending the hydrological record by ~70 years significantly increases the sample of historic drought events from which to conduct such research. Furthermore, the modelled data may be used to extend streamflow records used in seasonal hydrological forecasting with a hydrological analogues method (e.g. Svensson, 2016). The model calibrations may be applied to studies of the impacts of climate change on future hydrological extremes in the UK, such as in the Future Flows Hydrology project (Haxton et al., 2012), the outputs of which have been widely applied by water resources managers. The modelling framework developed in this study could extend the Future Flows Hydrology research using the more recent UKCP18 data (Met Office Hadley Centre, 2018). However, as with the Future Flows Hydrology project, users will need to be aware of the implications of the lack of artificial influence processes in the model.

## 7    Conclusions

In this paper, a novel multi-objective calibration method was derived and tested for 303 catchments in the UK, and the calibrations were used to reconstruct river flows back to 1891. The GR4J model was applied and calibrated using Latin Hypercube Sampling (LHS) and six evaluation metrics simultaneously to allow for the evaluation of high, median and low flows, thus optimising the calibrations for a wide range of potential applications. A best run (LHS1) and 500 model parameterisations (LHS500) were used to assess model uncertainty. Overall, the multi-objective calibration procedure has yielded excellent model results when compared to the observations, with the exception of only a few catchments. The reconstructed flows were appraised over 30 year moving windows, and were shown to provide good simulations of flow in the early parts of the record, where observations were available. Model performance and uncertainty during drought events was explored in nine case study catchments, and varied by catchment. The model simulations were used to derive the Standardised Streamflow Index, which allowed for an assessment of the model's ability to simulate significant deviations from a catchment's

"norm". The results showed that, despite observations regularly sitting outside the range of the LHS500, the peaks and troughs of the timeseries were well represented. Drought event accumulated deficits were extracted from the SSI data and the results were overall very good, demonstrating that the data from these model calibrations are suitable for the identification and characterisation of hydrological drought events in the UK.

The contributions of this paper are threefold: Firstly, the multi-objective model calibration framework applied here has been shown to provide robust model calibrations that can be applied in studies of both general and extreme hydrology. This framework could be applied elsewhere across Europe, and indeed globally to allow for spatially and temporally consistent simulations of hydrology with far reaching potential applications. Secondly, the model calibrations that have been derived for these 303 catchments in the UK can be used in further research and operational applications, such as for seasonal hydrological forecasting, or for assessing changes in river flows under climate change. Finally, this study has produced a crucial dataset of ~125 years of seamless flow reconstructions across the UK that will allow for the spatial and temporal investigation and quantification of past drought events, as well as long term trends in flows, that have never before been possible. These methods and results can provide a valuable step forward in our ability to plan for and forecast the onset, duration and termination of drought events in the UK, and overseas.

## 8 Acknowledgements

This research is an outcome of two projects, funded by the Natural Environment Research Council Drought and Water Scarcity Programme: Analysis of Historic Droughts and Water Scarcity in the UK [grant number NE/L01016X/1]; and Improving Predictions of Drought for User Decision-Making [IMPETUS, grant number NE/L010267/1]. The authors would also like to thank the developers of the airGR model at IRSTEA for their guidance.

## 9 Data Availability

Potential Evapotranspiration data: The PET dataset used in this study is freely available on the Environmental Information Data Centre (Tanguy et al., 2017).

Observed river flow data: Observed flow data was accessed via the National River Flow Archive, which provides daily and peak river flows for the UK for over 1500 gauging stations. (https://nrfa.ceh.ac.uk/)

Reconstructed flow data: The flow reconstructions produced in this study are freely available on the Environmental Information Data Centre (EIDC, Smith et al., 2018a) along with associated metadata on the models performance. The LHS1 and LHS500 model runs are provided separately within the EIDC dataset. The LHS1 files includes the deterministic simulation based on LHS1 parameter set, plus the upper and lower daily limits from the LHS500 to allow for the interpretation of the parameter uncertainty without the need to assess the full ensemble. It should be noted however that these upper and lower bounds cannot be implemented as timeseries in their own right as they do not represent individual ensemble members, and are instead comprised of multiple runs. The LHS500 files contain all 500 timeseries, and each catchment has a metadata file providing performance of each of the 500 runs for that catchment. The performance of the model in each catchment, as well as the reconstructed flow timeseries, can be explored using an interactive web application at https://shiny-apps.ceh.ac.uk/reconstruction_explorer/

Standardised Streamflow Index data: The SSI data derived from the LHS1 runs are also freely available from the Environmental Information Data Centre (Barker et al., 2018). This SSI data, along with further event analyses can be explored using an interactive web application at https://shiny-apps.ceh.ac.uk/hydro_drought_explorer/

## 10 Author Contributions

TPL and MT provided input data for the modelling. CP, SP, JH and SH assisted KAS in designing the modelling framework. KAS calibrated the model, ran the reconstructions and produced the plots. LJB calculated the SSI data and extracted drought events from the reconstructions. All authors contributed to the production of the manuscript, and its revisions.

## 11 Competing Interests

The authors declare that they have no conflict of interest.

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
