# Peer review of "A Multi-Objective Ensemble Approach to Hydrological Modelling in the UK: An Application to Historic Drought Reconstruction"

_Hydrology and Earth System Sciences, 2019_

## Referee Comment (RC1) · Anonymous Referee #1 · 6 Mar 2019

**General Comments**

This manuscript is a thorough evaluation and description of a new modeled dataset reconstructing historical flows in the UK. The authors do a good job outlining both the utility and limitations of the dataset they have created. This article makes very good use of graphics to convey complex information about a large number of data points; I especially like Figure 2. Overall, this is a high-quality paper, with just a few areas that require clarification (see "Specific Comments") or technical corrections (see below).

**Specific Comments**

Lines 358-360: The statement about selecting a "best" simulation rather than extracting

a mean or median from the ensemble appears to be a very strong statement based only on some qualitative examples. The authors could just say that selecting a "best" simulation is SOMETIMES more accurate than using an ensemble mean. Otherwise, if the authors wish to back up their statement, I think they would need to do a more thorough analysis comparing both LHS1 and the ensemble means (or medians) to the observations.

Lines 477-479: I don't quite follow the meaning of the sentence "They concluded that ... eliminate the influence of different PET inputs on runoff simulation." Does this mean that PET is not an important variable in predicting runoff? Does it mean that the hydrologic models have low sensitivity to small errors in PET? Please clarify.

**Technical Corrections**

Lines 70-73: These sentences are a little confusing, because it is unclear whether you mean the same thing by "hydrological models" and "rainfall-runoff models." Are you saying that your methods are different from those used by Caillouet et al (2017) in France, or that Caillouet et al (2017) is a rare example of the type of analysis you have done for the UK?

Lines 75-6: "They can be used ... prior to observational network" is an incomplete sentence. Please revise.

Line 125: It is not necessary to state that the catchments are shown in Figure 1, as this was already stated on line 123.

Line 193: Please also define "LHS500" in the methods section before using it here. At present, it is not defined until line 212.

Line 225: Please provide more information about what the Tweedie distribution is.

Line 323: Add an apostrophe at the end of "models."

Line 518: Change "catchments" to "catchment's"

---

## Referee Comment (RC2) · Anonymous Referee #2 · 10 Mar 2019

This paper uses a multi-objective approach to calibrate a fairly simple hydrologic model to predict discharge at a large number of catchments in the UK based on precipitation and temperature observations. The stated purpose of the exercise was to hindcast streamflow during historical early 20th century droughts that occured prior to the systematic collection of discharge observations on UK streams, but (crucially) not before available meteorological records. The results show that the relatively simple hydrologic model that was used (4 parameters) was able to capture streamflow variability well, over the wide range of catchments included in the survey. The study showed little evidence of non-stationarity in parameter calibration, which allowed historical droughts to be hincasted with a decent level of confidence.

[Figure]

Major remarks The study is methodologically solid. The paper is well written and methods and results are described clearly and in sufficient details. However, I am not sure I understand the contribution of the paper beyond a solid regional study of UK streams. This is without a doubt a useful practical contribution for the UK water resources community, but you should do a better job at discussing general implications of the research in the introduction and discussion. To be excessively blunt, as a scientist that has no particular interest in UK streams (like a large chunk of HESS readership), why should I care? To be a bit more specific, you explicitly lists the intended contributions of the paper in the conclusions (L527). At face value, these contributions are sufficiently general to interest non-UK readers and should be stated upfront (the intro is very much UK specific currently). However, I think that these arguments currently lack substance and should be further developed:

1. You mention your multi-objective calibration approach as the first general contribution of the paper. As you admit yourself (L91), the concept itself of multi-objective calibration is not new and the section where you describe model selection (3.4) is particularly cryptic. If multi-objective calibration is indeed a key contribution of the paper, please describe the approach specifically (How are the model parametrizations "ranked"? How are each of the criteria weighted to come up with a composite ranking?) and spell out clearly what the novelty is compared to existing approaches.

2. Second, you claim that the approach can be used not only to hindcast droughts but also to predict catchment responses to future climate change. In order to make such a claim, you ought to address th elephant in the room, which is that your approach does not accomodate non-stationarities in the calibrated parameters (e.g., related to land use change and human adaptation). Your result suggest that these factor were not much of a problem for historical simulations (except for heavily altered catchment), but if there is one thing that climate studies tell us is that the past is not necessarily representative of the future. I do agree that your results are interesting and can be leveraged to study the hydrological impacts of climate change, but the implied caveats
— and potential avenues to go around them — should be discussed. I am specifically thinking of the potential to leverage satellite observations of land use change and/or modules integrating human adaptation to large scale hydrological models (e.g, Bierkens 2015, Calvin 2018).

3.Third, you argue that the study provides important spatio-temporal data on historical drought in the UK (so far so good) which can be used to plan and forecast the onset, duration and termination of drought events in the UK and overseas. First off, it is not clear to me how, specifically, how the historical reanalysis you describe can be used to forecast and mitigate the effect of future droughts (see previous point) – if you have a specific idea here, please make it explicit. Most importantly, your method relies on the fact that a large volume of high quality meteorological observations (for both P and PET) were available in the early 20th century, before river discharges were systematically gauged. This was definitely the case for the UK, but in order to argue that the approach you propose is applicable beyond the UK (which would make it more relevant to the global hydrologic community), you have to show that what happened in the UK is not an exception. It can very well be that met data was collected way before flow data in other countries as well, but you have to make this argument explicit (and ideally back it up with some data).

Minor comments

L210 I am not sure I understand your multi-objective approach to select catchments. How do you weigh different criteria when ranking the parametrization (e.g., how do you differentiate a parametrization A with a NSE of 0.64 and a Q95APE of 34 from a parametrization B with a NSE of 0.70 and a Q95 APE of 40 – which one dominates?). What optimality concept is your approach consistent with (pareto, maxi-min (i..e maximizing the worst performing metrics), . . .)

There are lots of accronyms to remember. A Table summarizing the abbreviations would be useful

Fig 5: labeled pointers showing the catchemnt that you specifically discuss in the text would be useful.

L132, 502: Please refrain from citing work in preparation.

Refs Calvin, Katherine, and Ben Bond-Lamberty. "Integrated human-earth system modeling—state of the science and future directions." Environmental Research Letters 13.6 (2018): 063006. Bierkens, Marc FP. "Global hydrology 2015: State, trends, and directions." Water Resources Research 51.7 (2015): 4923-4947.

---

## Referee Comment (RC3) · James Stagge (Referee) · 22 Mar 2019

Here the authors address two unique research questions. First, the authors define a multi-objective approach to calibrating a hydrologic model to consider low flows, high flows, and water balance. Second, they use this approach to reconstruct flows for rivers throughout the UK beginning in the 1891, made possible by recovered meteorologic datasets.

The paper is well-written, of strong interests for HESS readers and a novel piece of research. I have some concerns about a general lack of reference to the hydrologic calibration literature, particularly with relation to prior multi-objective approaches. The

authors' application is certainly novel and they made choices to weight their multiple objectives a priori, which is a realistic approach when repeating this for many watersheds. However, there are more advanced multi-objective schemes that should be mentioned for context (and potentially for follow-up research). Because of this weighting approach, there must be some discussion of how the objectives are related to one another and how these weightings affect results.

Overall, I recommend this article for publication pending the major revisions to provide a better literature context and to better explain the objective weighting scheme's effects.

Major Comments 1. I have a concern that there is a wide body of calibration/optimization literature not being referenced in this paper. Many approaches have been used for hydrologic model parameter calibration, and although the paper mentions some, there are gaps that could put this work in context. I suggest to at least mention PEST, which is a single objective optimization scheme, but almost ubiquitous in the U.S. hydrologic community. Wallner (2012) "Evaluation of different calibration strategies for large scale continuous hydrological modelling" provides a good overview of these calibration strategies.

2. Although the words "multi-objective optimization" aren't often written together in the text, this approach appears to be an a priori multi-objective optimization. By using the sum of each objective's rank as your objective, you have defined weightings a priori to merge multiple objectives into a single objective function. Please include at least one or two sentences explaining this and mentioning the difference between this and a posteriori multi-objective optimization (below).

I mention this because you state that "multi-objective optimization methods have been advancing since the turn of the century", but this area has a pretty rich literature that goes back well into the 1990s. Additionally, most optimization researchers think of a posteriori (not a priori) when they think of multi-objective optimization. A posteriori approaches try to find a set of non-dominated Pareto optimal solutions and then select

[Figure]

the best compromise afterwards. You might include references to other multi-objective papers that take this approach like:

"Multiobjective Automatic Parameter Calibration of a Hydrological Model" (Jung et al, 2017) "Comparing multi-objective optimization techniques to calibrate a conceptual hydrological model using in situ runoff and daily GRACE data" (Mostafaie et al. 2018) "Automatic calibration of HEC-HMS using single-objective and multi-objective PSO algorithms" (Kamali et al. 2013) "Multi-objective calibration of a distributed hydrological model (WetSpa) using a genetic algorithm" (Shafi and de Smedt 2009)

Or consider some of their references for older publications.

3. Because of the a priori weighting (Comment #2), please provide information about how the multiple objectives are related to one another. Are some highly correlated? Negatively correlated? If, for instance, the rankings from the 4 high/water balance objectives operate as one and the 2 low flow indices operate as one, is there a concern that you are overweighting towards high flows?

4. Line 245-250: I find it surprising that there is a single very poor fit among nearly perfect fits, for example in Cornwall. As you are mentioning the reasons for poor fits in this paragraph, it is important to mention there does not appear to be a spatial pattern. Presumably, the same abstractions and groundwater issues affect the 0-10% threshold poor fit as its > 90% good fit neighbors. Are there any other feasible explanations?

Minor Comments

Line 45 – Suggest 1 or 2 more references to fill out the discussion of low flow climate projections for the UK.

Line 70 – You may want to mention some proxy-based reconstructions; for example Jones et al (1984) "Riverflow reconstruction from tree rings in southern Britain" or the Old World Drought Atlas (Cook et al 2015) "Old World megadroughts and pluvials during the Common Era" which covers the UK.

Line 193 – Please define LHS500. This is the first time it is included in the text (only in the abstract).

Table 2 – If possible, please try to fit the ranges on a single line of this table.

Lines 273 – You do a great job of describing a low UncW and low ContR as biased and under-sensitive - this is a helpful translation for readers. As a reader, I would also like a description of the converse. What does high UncW and high ContR mean?

Line 344 - Can you provide a description of which objective function(s) is driving the best fit parameter set in the Avon to consistently overestimate low flows?

Line 372 – Please add the words "we consider" before "SSI values. . .". The thresholds of -1 and -1.5 are largely arbitrary and more of a convention than a true definition.

Figure 9 – Please mention that you are plotting the mid-point of each event in the caption. It is currently only in the text (Line 417).

Figure 9 – For the Crimple watershed, there are 3 unique drought events for the Modeled data shown in the period 1975-1979. But Figure 8 shows only 2 crosses of the -1 threshold. Please confirm what is going on here.

---

## Author Comment (AC1) · 29 Apr 2019

Authors Response to Anonymous Referee 1

The authors responses are given in a bold typeface.

**General Comments**

This manuscript is a thorough evaluation and description of a new modeled dataset reconstructing historical flows in the UK. The authors do a good job outlining both the utility and limitations of the dataset they have created. This article makes very good use of graphics to convey complex information about a large number of data points; I

especially like Figure 2. Overall, this is a high-quality paper, with just a few areas that require clarification (see "Specific Comments") or technical corrections (see below).

**RESPONSE: We thank the reviewer for their kind words on the manuscript, and are pleased they valued our use of graphics.**

**Specific Comments**

Lines 358-360: The statement about selecting a "best" simulation rather than extracting a mean or median from the ensemble appears to be a very strong statement based only on some qualitative examples. The authors could just say that selecting a "best" simulation is SOMETIMES more accurate than using an ensemble mean. Otherwise, if the authors wish to back up their statement, I think they would need to do a more thorough analysis comparing both LHS1 and the ensemble means (or medians) to the observations.

**RESPONSE: Thank you for this comment, we agree completely and will amend the manuscript to indicate that this is a qualitative and possibly case specific statement. E.g. "This indicates that selecting the "best" simulation where a deterministic result is needed is more appropriate, in these cases, than extracting a mean or median from the ensemble."**

Lines 477-479: I don't quite follow the meaning of the sentence "They concluded that . . . eliminate the influence of different PET inputs on runoff simulation." Does this mean that PET is not an important variable in predicting runoff? Does it mean that the hydrologic models have low sensitivity to small errors in PET? Please clarify.

**RESPONSE: This statement implies that the calibration of a hydrological model can eliminate some of the uncertainties that may be derived from the quality of the PET data. PET is a very important variable in predicting runoff, but using poorer quality temperature data PET instead of very high spatial and temporal resolution data is unlikely to significantly affect the streamflow output, as the**

**calibration of the hydrological model can implicitly account for such errors. The authors will amend these few sentences to be clearer. E.g. "Tanguy et al. (2018) considered the impact of poorer quality and lower density of temperature data on the derivation of the PET dataset that was employed in this study and concluded that, whilst PET is an important variable for predicting runoff, the influence of degraded PET input that result from low quality temperature data on runoff simulation can be limited by the adequate calibration of hydrological models (Bai et al., 2016; Seiller and Anctil, 2016). Thus, the Tanguy et al. (2018) PET dataset is considered suitable for use in hydrological models, especially if they are calibrated to this dataset."**

**Technical Corrections**

Lines 70-73: These sentences are a little confusing, because it is unclear whether you mean the same thing by "hydrological models" and "rainfall-runoff models." Are you saying that your methods are different from those used by Caillouet et al (2017) in France, or that Caillouet et al (2017) is a rare example of the type of analysis you have done for the UK?

**RESPONSE: Yes, we mean the same thing by hydrological and rainfall-runoff models. We have replaced references to "rainfall-runoff models" with "hydrological models" for consistency. We mean the latter – That studies such as this, and Caillouet et al (2017), using meteorological data with hydrological models, are rare; Caillouet et al also used a hydrological model to reconstruct flows, but: they used reanalysis data as climate input data where we have used observed data; our calibration and uncertainty analyses methods are different; and our drought event extraction techniques also differ.**

Lines 75-6: "They can be used . . . prior to observational network" is an incomplete sentence. Please revise.

**RESPONSE: We consider this a complete sentence, but have amended it for clar-**

[Figure]

**ity: "They can be used to extend flow records back in time, creating very long sequences that extend back beyond the initiation of the observational network"**

Line 125: It is not necessary to state that the catchments are shown in Figure 1, as this was already stated on line 123.

**RESPONSE: We have removed this sentence.**

Line 193: Please also define "LHS500" in the methods section before using it here. At present, it is not defined until line 212.

**RESPONSE: The Sentence has been corrected to: "The upper and lower daily limits of the 500 top ranking parameterisations (see Section 3.4 for details on the ranking process) were used to calculate. . ."**

Line 225: Please provide more information about what the Tweedie distribution is.

**RESPONSE: Readers may refer to the Svensson et al paper if they are further interested in the distribution, however, the final sentence of this paragraph has been amended to state: "The Tweedie distribution, which is a flexible three-parameter distribution that has a lower bound at zero, has been shown to perform effectively for UK river flows, across a wide range of near-natural Benchmark catchments (Svensson et al., 2017)."**

Line 323: Add an apostrophe at the end of "models."

**RESPONSE: This has been added**

Line 518: Change "catchments" to "catchment's"

**RESPONSE: This has been added**

Line 523: Change "This contributions" to "The contributions"

**RESPONSE: This has been corrected**

**We thank the reviewer for highlighting these technical errors.**

---

## Author Response (AR1)

**Authors Response to Anonymous Referee #1**

**General Comments**

**Reviewer Comment:**

This manuscript is a thorough evaluation and description of a new modeled dataset reconstructing historical flows in the UK. The authors do a good job outlining both the utility and limitations of the dataset they have created. This article makes very good use of graphics to convey complex information about a large number of data points; I especially like Figure 2. Overall, this is a high-quality paper, with just a few areas that require clarification (see "Specific Comments") or technical corrections (see below).

*Authors Initial Comments:*

*We thank the reviewer for their kind words on the manuscript, and are pleased they valued our use of graphics.*

**Specific Comments**

**Reviewer Comment:**

Lines 358-360: The statement about selecting a "best" simulation rather than extracting a mean or median from the ensemble appears to be a very strong statement based only on some qualitative examples. The authors could just say that selecting a "best" simulation is SOMETIMES more accurate than using an ensemble mean. Otherwise, if the authors wish to back up their statement, I think they would need to do a more thorough analysis comparing both LHS1 and the ensemble means (or medians) to the observations.

*Authors Initial Comments:*

*Thank you for this comment, we agree completely and will amend the manuscript to indicate that this is a qualitative and possibly case specific statement. E.g. "This indicates that selecting the "best" simulation where a deterministic result is needed is more appropriate, in these cases, than extracting a mean or median from the ensemble."*

**Reviewer Comment:**

Lines 477-479: I don't quite follow the meaning of the sentence "They concluded that . . . eliminate the influence of different PET inputs on runoff simulation." Does this mean that PET is not an important variable in predicting runoff? Does it mean that the hydrologic models have low sensitivity to small errors in PET? Please clarify.

*Authors Initial Comments:*

*This statement implies that the calibration of a hydrological model can eliminate some of the uncertainties that may be derived from the quality of the PET data. PET is a very important variable in predicting runoff, but using poorer quality temperature data PET instead of very high spatial and temporal resolution data is unlikely to significantly affect the streamflow output, as the calibration of the hydrological model can implicitly account for such errors. The authors will amend these few sentences to be clearer. E.g. Tanguy et al. (2018) considered the impact of poorer quality and lower density of temperature data on the derivation of the PET dataset that was employed in this study and concluded that, whilst PET is an important variable for predicting runoff, the influence of degraded PET input that result from low quality temperature data on runoff simulation can be limited by the adequate calibration of hydrological models (Bai et al., 2016; Seiller and Anctil, 2016). Thus, the Tanguy et al. (2018) PET dataset is considered suitable for use in hydrological models, especially if they are calibrated to this dataset.*

**Technical Corrections**

**Reviewer Comment:**

Lines 70-73: These sentences are a little confusing, because it is unclear whether you mean the same thing by "hydrological models" and "rainfall-runoff models." Are you saying that your methods are different from those used by Caillouet et al (2017) in France, or that Caillouet et al (2017) is a rare example of the type of analysis you have done for the UK?

*Authors Initial Comments:*

*Yes, we mean the same thing by hydrological and rainfall-runoff models. We have replaced references to "rainfall-runoff models" with "hydrological models" for consistency. We mean the latter – That studies such as this, and Caillouet et al (2017), using meteorological data with hydrological models, are rare; Caillouet et al also used a hydrological model to reconstruct flows, but: they used reanalysis data as climate input data where we have used observed data; our calibration and uncertainty analyses methods are different; and our drought event extraction techniques also differ.*

**Reviewer Comment:**

Lines 75-6: "They can be used . . . prior to observational network" is an incomplete sentence. Please revise.

*Authors Initial Comments:*

*We consider this a complete sentence, but have amended it for clarity: "They can be used to extend flow records back in time, creating very long sequences that extend back beyond the initiation of the observational network"*

**Reviewer Comment:**

Line 125: It is not necessary to state that the catchments are shown in Figure 1, as this was already stated on line 123.

*Authors Initial Comments:*

*We have removed this sentence.*

**Reviewer Comment:**

Line 193: Please also define "LHS500" in the methods section before using it here. At present, it is not defined until line 212.

*Authors Initial Comments:*

*The Sentence has been corrected to: "The upper and lower daily limits of the 500 top ranking parameterisations (see Section* **Error! Reference source not found.** *for details on the ranking process) were used to calculate…"*

**Reviewer Comment:**

Line 225: Please provide more information about what the Tweedie distribution is.

*Authors Initial Comments:*

*Readers may refer to the Svensson et al paper if they are further interested in the distribution, however, the final sentence of this paragraph has been amended to state: "The Tweedie distribution, which is a flexible three-parameter distribution that has a lower bound at zero, has been shown to perform effectively for UK river flows, across a wide range of near-natural Benchmark catchments (Svensson et al., 2017)."*

**Reviewer Comment:**

Line 323: Add an apostrophe at the end of "models."

*Authors Initial Comments:*

*This has been added*

**Reviewer Comment:**

Line 518: Change "catchments" to "catchment's"

*Authors Initial Comments:*

75 *This has been added*

**Reviewer Comment:**

Line 523: Change "This contributions" to "The contributions"

*Authors Initial Comments:*

*This has been corrected*

80

*We thank the reviewer for highlighting these technical errors.*

*Authors Initial Comments:*

*This has been added*

Line 523: Change "This contributions" to "The contributions"

85 **Reviewer Comment:**

This paper uses a multi-objective approach to calibrate a fairly simple hydrologic model to predict discharge at a large number of catchments in the UK based on precipitation and temperature observations. The stated purpose of the exercise was to hindcast streamflow during historical early 20th century droughts that occurred prior to the systematic collection of discharge observations on UK streams, but (crucially) not before available meteorological records. The results show that the relatively

90 simple hydrologic model that was used (4 parameters) was able to capture streamflow variability well, over the wide range of catchments included in the survey. The study showed little evidence of non-stationarity in parameter calibration, which allowed historical droughts to be hindcasted with a decent level of confidence.

**Major remarks**

**Reviewer Comment:**

95 The study is methodologically solid. The paper is well written and methods and results are described clearly and in sufficient details. However, I am not sure I understand the contribution of the paper beyond a solid regional study of UK streams. This is without a doubt a useful practical contribution for the UK water resources community, but you should do a better job at discussing general implications of the research in the introduction and discussion. To be excessively blunt, as a scientist that has no particular interest in UK streams (like a large chunk of HESS readership), why should I care? To be a bit more specific,

100 you explicitly lists the intended contributions of the paper in the conclusions (L527). At face value, these contributions are sufficiently general to interest non-UK readers and should be stated upfront (the intro is very much UK specific currently). However, I think that these arguments currently lack substance and should be further developed:

*Authors Initial Comments:*

*We thank you for your comments, and appreciate that the introduction could be better framed. We believe that the methods*
105 *employed in this study are applicable elsewhere across the globe, as well as in time. The multi-objective approach to model calibration used here is not exclusive to the UK, nor to reconstructions, but may also be used to calibrate models elsewhere for flow forecasting and longer term projections. Similarly, it could, with sufficient computational resources, be applied to more complex hydrological models. Furthermore, we believe that the data produced from this research will be of wider interest in the framing of historic flows and extreme events from a European perspective. If you agree that the contributions outlined*
110 *in the discussion are of sufficient interest to wider readers, we will revise the manuscript to make these points clearer in the abstract and the introduction.*

*Authors Final Comments:*

*The first paragraph detailing UK drought events has been cut down/removed. European literature has been added to the climate projections statement. Detail on GRDC global network evolution has been added. Detailed examples of qualitative*
115 *past UK droughts have been generalised, and other European drought studies using documentary evidence have been cited. Global met data availability from CRU has been cited (New et al, 2000). The aims have been generalised with fewer references to spatial location.*

*The last few sentences of the abstract have been slightly amended to read: "This paper provides three key contributions: 1) an robust multi-objective model calibration framework for calibrating catchment models for use in both general and extreme*
120 *hydrology; 2) model calibrations for the 303 UK catchments that could be used in further research, and operational applications such as hydrological forecasting; and 3) ~125 years of spatially and temporally consistent reconstructed flow data derived that will allow comprehensive quantitative assessments of past UK drought events, as well as long term analyses*

*of hydrological variability that have not been previously possible, thus enabling water resource managers to better plan for extreme events, and build more resilient systems for the future."*

**Reviewer Comment:**

1. You mention your multi-objective calibration approach as the first general contribution of the paper. As you admit yourself (L91), the concept itself of multi-objective calibration is not new and the section where you describe model selection (3.4) is particularly cryptic. If multi-objective calibration is indeed a key contribution of the paper, please describe the approach specifically (How are the model parametrizations "ranked"? How are each of the criteria weighted to come up with a composite ranking?) and spell out clearly what the novelty is compared to existing approaches.

*Authors Initial Comments:*

*We apologise that the method has not been clearly set out, and that you found section 3.4 cryptic; we will endeavour to make it more transparent. We will likely include the code that was used for the ranking process in the supplementary materials for the readers' reference. The third reviewer has also commented that we need to put our method in the context of existing multi-objective calibration approaches, so we will make sure this issue is addressed in the revised manuscript.*

*Authors Final Comments:*

*We amended sections 3.2, 3.3, and 3.4 to be one section "Calibration Strategy" (with subsections) in order to allow a space for general comment on the approach and how it fits with previous research. Text added "The GR4J model was calibrated for this study incorporating concepts from GLUE type Bayesian approaches (Beven and Freer, 2001), and multi-objective Pareto-optimal solutions (Yapo et al., 1998). The approach consisted of three stages, the details of which are further elaborated in this sub-section: firstly, the feasible parameter space was determined, and sampled using Latin Hypercube Sampling (LHS) (McKay et al., 1979); secondly the model was run, and six evaluation metrics were calculated for each parameter set; and thirdly the top 500 parameter sets for each catchment were selected using a very simple Pareto-optimising ranking approach, accounting for non-acceptable trade-offs (Efstratiadis and Koutsoyiannis, 2010)."*

*The section on ranking has been re-written as:*

*"In order to optimise six evaluation metrics, the 500,000 model parameterisations were ranked from best to worst by their scores for each metric in turn, and these ranks were then summed to create a total rank. This total, or "basic", rank was used to reorder the parameterisations for each catchment from best to worst, accounting for all metrics. However, the scores of the 500,000 model parameterisations were not normally distributed, and it was found that unacceptable trade-offs between metrics were occurring, whereby nominal increases in one metric were taking preference over quite significant decreases in other metrics. Therefore, a series of thresholds of acceptability were set, as shown in Table 3. A simple iterative search algorithm was then used to re-rank the list according to these thresholds, whilst retaining their original ranks within each threshold group. For example, if the first, third and fourth parameterisations in the basic rank met the hardest threshold for all six metrics, but the second ranked parameterisation did not, they would be bumped up the rankings, above the second resulting in a list of [1, 3, 4, 2...]. All parameterisations meeting the hardest thresholds were prioritised before the algorithm switched to search for those in meeting the middle thresholds, and so on. From this final list, the top ranking optimum parameter set was extracted for deterministic model applications, herein referred to as LHS1. Due to the variability of the performance across catchments, where hundreds of thousands of parameter sets met the hardest threshold in some catchments, whilst none met even the softest threshold in other catchments, it was decided that a 'limit of acceptability' approach after Beven (2006) would not be appropriate. Therefore, a proportion of the sampled model parameterisations, the top 500 (herein referred to as LHS500), were taken forward to provide an indication of parameter uncertainty within the flow simulations. The extent to which the threshold re-ranking influenced the rankings varied by catchment due to the differences in mode*

*performance. Figure 2 shows the NSE and logNSE scores of the 500,000 model parameterisations (though this graph has been limited to show only those with positive scores for both metrics) for the River Greet in Severn Trent Region. This figure demonstrates how the basic ranking system identified 500 parameterisations close to the Pareto front of NSE vs logNSE, however parameterisations with scores that were lower for NSE than logNSE were selected. By applying the thresholds, parameterisations with an NSE lower than 0.4 were rejected, and replaced with others within the acceptable range for all metrics"*

*An illustrative figure has been added to the manuscript.*

**Reviewer Comment:**

2. Second, you claim that the approach can be used not only to hindcast droughts but also to predict catchment responses to future climate change. In order to make such a claim, you ought to address the elephant in the room, which is that your approach does not accommodate non-stationarities in the calibrated parameters (e.g., related to land use change and human adaptation). Your result suggest that these factor were not much of a problem for historical simulations (except for heavily altered catchment), but if there is one thing that climate studies tell us is that the past is not necessarily representative of the future. I do agree that your results are interesting and can be leveraged to study the hydrological impacts of climate change, but the implied caveats and potential avenues to go around them should be discussed. I am specifically thinking of the potential to leverage satellite observations of land use change and/or modules integrating human adaptation to large scale hydrological models (e.g, Bierkens 2015, Calvin 2018).

*Authors Initial Comments:*

*We agree that land use changes and human adaptations are likely to influence flows significantly in the context of climate change projections. However, we are reassured by the integrity of the model results when compared to the longer observed time series. Previous modelling studies have used lumped catchment models to simulate flows under climate change (e.g. future flows hydrology, Haxton et al 2012), and the results have been widely employed in water resource management simulations. We anticipate that this modelling framework, applied to more recent climate projections such as UKCP18 may be equally useful for decision makers, especially in the near-natural low flows benchmark network catchments, where water resource managers may use the flow projections to assess water availability, and subsequently run the flow projections through water resource models to simulate the impacts of changes in human influence over time. We discuss the lack of human influence in the model in the discussion section, but we will add this caveat to the mention of future applications, and also reference the Future Flows Hydrology study in the manuscript.*

*Haxton, T.; Crooks, S.; Jackson, C.R.; Barkwith, A.K.A.P.; Kelvin, J.; Williamson, J.; Mackay, J.D.; Wang, L.; Davies, H.; Young, A.; Prudhomme, C. (2012). Future flows hydrology data. NERC Environmental Information Data Centre. https://doi.org/10.5285/f3723162-4fed-4d9d-92c6-dd17412fa37b*

*Authors Final Comments:*

*The following sentences have been added to this section of the discussion, citing your suggested literature: "Human interactions are a common problem in hydrological modelling that remain largely understudied (Calvin and Bond-Lamberty, 2018). Whilst global scale models have been advancing in socio-hydrology, making use of satellite information and governmental estimates of total water consumption, the data to support such endeavours is lacking (Bierkens, 2015). Small scale catchment models would need to rely on significant amounts of abstraction and licencing data as well as reservoir operation procedures, the details of which are often sensitive and/or unavailable."*

*The following sentences have been added to the final paragraph of the discussion: "The model calibrations may be applied to studies of the impacts of climate change on future hydrological extremes in the UK, such as in the Future Flows Hydrology project (Haxton et al., 2012), the data from which has been widely applied by water resources managers. The modelling framework developed in this study could extend the Future Flows Hydrology research using the more recent UKCP18 data (Met Office Hadley Centre, 2018). However, as with the Future Flows Hydrology project, users will need to be aware of the implications of the lack of artificial influence processes in the model."*

**Reviewer Comment:**

3. Third, you argue that the study provides important spatio-temporal data on historical drought in the UK (so far so good) which can be used to plan and forecast the onset, duration and termination of drought events in the UK and overseas. First off, it is not clear to me how, specifically, how the historical reanalysis you describe can be used to forecast and mitigate the effect of future droughts (see previous point) – if you have a specific idea here, please make it explicit.

*Authors Initial Comments:*

*Historical data can provide vital context when faced with an ongoing drought episode. Whilst, as you say, the past may not necessarily be representative of the future, using ensembles of historical drought events can gain insight into the probabilities of the termination of a current event over a certain time period (e.g. Parry et al, 2018). Knowledge of historic events can also be used to explore statistical correlations with atmospheric drivers of droughts that may help predict the onset of events (e.g. Lavers et al, 2015). In these approaches, extending the hydrological record by ~70 years significantly increases the sample of historic drought events from which to conduct such research. Furthermore, the modelled data may be used to extend streamflow records used in seasonal hydrological forecasting with a hydrological analogues method (e.g. Svensson, 2016), and the model set-up is already being applied in seasonal forecasting using an Ensemble Streamflow Prediction approach in the UK Hydrological Outlooks ([www.hydoutuk.net](www.hydoutuk.net)). This will also be added to the manuscript.*

*Parry, S., Wilby, R., Prudhomme, C., Wood, P., McKenzie, A. (2018) Demonstrating the utility of a drought termination framework: prospects for groundwater level recovery in England and Wales in 2018 or beyond. Environmental Research Letters.*

*Lavers, D., Hannah, D., Bradley, C., (2015) Connecting large-scale atmospheric circulation, river flow and groundwater levels in a chalk catchment in southern England. Journal of Hydrology 523, 179-189.*

*Svensson, C. (2016) Seasonal river flow forecasts for the United Kingdom using persistence and historical analogues. Hydrological Sciences Journal. 61 (1), 19-35.*

*Authors Final Comments:*

*The following sentences have been added to the final paragraph of the discussion: "Ensembles of historical drought events can be used to provide insight into the probabilities of the termination of a current event over a certain time period (e.g. Parry et al., 2018). Knowledge of historic events can also be used to explore statistical correlations with atmospheric drivers of droughts that may help predict the onset of events (e.g. Lavers et al., 2015). In these approaches, extending the hydrological record by ~70 years significantly increases the sample of historic drought events from which to conduct such research. Furthermore, the modelled data may be used to extend streamflow records used in seasonal hydrological forecasting with a hydrological analogues method (e.g. Svensson, 2016)."*

**Reviewer Comment:**

Most importantly, your method relies on the fact that a large volume of high quality meteorological observations (for both P and PET) were available in the early 20th century, before river discharges were systematically gauged. This was definitely the case for the UK, but in order to argue that the approach you propose is applicable beyond the UK (which would make it more

relevant to the global hydrologic community), you have to show that what happened in the UK is not an exception. It can very well be that met data was collected way before flow data in other countries as well, but you have to make this argument explicit (and ideally back it up with some data).

*Authors Initial Comments:*

*We believe that it is common that met data records begin before hydrological data records (within Europe at least), simply due to the relative complexities of recording temperature and rainfall over river levels or flows. Newly digitised observed climate datasets (such as the one employed in this study) are becoming increasingly extending observed series held by met services across Europe. Furthermore, Caillouet et al (2017) made use of modelled climate reanalysis data, and the approach could also be applied to other long term reconstructed climate datasets (such as the monthly Casty et al 2007 data). This comment will be added to the manuscript.*

*Caillouet, L., Vidal, J. P., Sauquet, E., Devers, A., and Graff, B.: Ensemble reconstruction of spatio-temporal extreme low-flow events in France since 1871, Hydrol. Earth Syst. Sci., 21, 2923-2951, 10.5194/hess-21-2923- 2017, 2017.*

*Casty, C., Raible, C. C., Stocker, T. F., Wanner, H., Luterbacher, J.: A European pattern climatology 1766-2000: Climate Dynamics, 29, 7-8, 791-805, 10.1007/s00382-007-0257-6, 2007.*

*Authors Final Comments:*

*Lines were added to the introduction: "Meteorological records of rainfall and temperature generally extend further back than hydrological data, often providing data from the turn of the 21$^{st}$ century (New et al., 2000), and occasionally as far back as the mid-20$^{th}$ century. Modelled climate reanalysis data (e.g. Compo et al., 2011), and long term reconstructed climate datasets (e.g. Casty et al., 2007) have been developed for use in scientific research, and can be fed into hydrological models to quantitatively reconstruct river flows beyond the limits of the observational period"*

**Minor comments**

**Reviewer Comment:**

L210 I am not sure I understand your multi-objective approach to select catchments. How do you weigh different criteria when ranking the parametrization (e.g., how do you differentiate a parametrization A with a NSE of 0.64 and a Q95APE of 34 from a parametrization B with a NSE of 0.70 and a Q95 APE of 40 – which one dominates?). What optimality concept is your approach consistent with (pareto, maxi-min (i..e maximizing the worst performing metrics), . . .)

*Authors Initial Comments:*

*The ranking was done as simply as possible, and does not conform to a traditional optimality concept due to the need to rank by 6 metrics at once. The matrix of 500,000 parameter sets and their scores was sorted first by NSE and a rank column was added giving each parameterisation a rank (1 best to 500,000 worst); the matrix was then sorted again but by logNSE and a new rank column was added; then again by absPBIAS etc. until there were 6 rank columns, one for each metric. The ranks were then summed, and the matrix was ordered by this total rank (with the lowest number being the best parameter set).*

*However, we found that this left us with a sub-optimal scoring system, as slight improvements in one metric were occasionally outweighing more severe degradations in other metrics, e.g. absPBIAS scores better by 0.001 but NSE scores worse by 0.1). This is why we then set the thresholds. We took the ranked matrix, and starting at the top, looked down the rows of parameterisations until we found one that met the hardest threshold criteria for all 6 metrics. If this was not the originally top ranking parameterisation, it was bumped to the top of the list, and the search was run again. If a second parameterisation was found to meet all 6 criteria, it was then bumped to second place, and the search was run again. Etc.*

*This created a matrix where all parameterisations that met the hardest criteria were at the top of the list (ordered by their original rankings), followed by those that met the middle criteria (ordered by their original rankings), followed by the softest etc.*

*This was done for each catchment individually.*

*As mentioned earlier, we will endeavour to clarify this in the revised manuscript, and will likely provide the R code.*

*Authors Final Comments:*

*As explained in response to major comment 1, we have made revisions to clarify this.*

**Reviewer Comment:**

There are lots of acronyms to remember. A Table summarizing the abbreviations would be useful

*Authors Initial Comments:*

*We will consult with the editors and include a table of acronyms in the supplementary information, if appropriate for the journal.*

*Authors Final Comments:*

*A table of acronyms has been added to the supplementary information*

**Reviewer Comment:**

Fig 5: labelled pointers showing the catchment that you specifically discuss in the text would be useful.

*Authors Initial Comments:*

*We will add markers to the figure*

*Authors Final Comments:*

*We have experimented with this and found that it complicated the plot significantly. We would prefer to leave it as it is, and have better described the points in the text, as:* "The Avon at Evesham in ST region, the Dee at Manley Hall in North West England North Wales (NWENW) region, and the Bedford Ouse catchment in Anglian (ANG) region, show reduced model performance earlier in the record, with the bars moving through orange and yellow shades as they stretch towards the centre of the circle". *"The Lee at Feildes Weir in SE region (plotted at the boundary with ANG region) also shows variation in performance across most metrics, although in this catchment, the performance is good (plotted in black) at the start and end of the record, with poorer performance (shown in yellow) around the start years of 1920-1940 (evaluation years of 1920 to 1970)." And "In contrast to this, the Dee at Woodend in East Scotland, and the Severn at Bewdley in Severn Trent region, which have the longest records in their regions, show more temporal stability in the model performances (with black colouring for the whole bar)."*

**Reviewer Comment:**

L132, 502: Please refrain from citing work in preparation.

*Authors Final Comments:*

*We have removed the Legg reference, the Barker reference has since been published in HESSD, so we have updated the reference.*

315 **Reviewer Comment:**

Here the authors address two unique research questions. First, the authors define a multi-objective approach to calibrating a hydrologic model to consider low flows, high flows, and water balance. Second, they use this approach to reconstruct flows for rivers throughout the UK beginning in the 1891, made possible by recovered meteorologic datasets.

The paper is well-written, of strong interests for HESS readers and a novel piece of research. I have some concerns about a

320 general lack of reference to the hydrologic calibration literature, particularly with relation to prior multi-objective approaches. The authors' application is certainly novel and they made choices to weight their multiple objectives a priori, which is a realistic approach when repeating this for many watersheds. However, there are more advanced multi-objective schemes that should be mentioned for context (and potentially for follow-up research). Because of this weighting approach, there must be some discussion of how the objectives are related to one another and how these weightings affect results.

325 Overall, I recommend this article for publication pending the major revisions to provide a better literature context and to better explain the objective weighting scheme's effects.

*Authors Initial Comments:*

*We thank you for your kind words Jim, and are glad that you deem the research novel and of strong interest to HESS readers. We appreciate your concern for the current lack of reference to the literature regarding multi-objective calibration procedures,*

330 *and will ensure that this is addressed in the revised manuscript.*

**Major Comments**

**Reviewer Comment:**

1. I have a concern that there is a wide body of calibration/optimization literature not being referenced in this paper. Many approaches have been used for hydrologic model parameter calibration, and although the paper mentions some, there are gaps

335 that could put this work in context. I suggest to at least mention PEST, which is a single objective optimization scheme, but almost ubiquitous in the U.S. hydrologic community. Wallner (2012) "Evaluation of different calibration strategies for large scale continuous hydrological modelling" provides a good overview of these calibration strategies.

*Authors Initial Comments:*

*Thank you for noticing this oversight, we will insert reference to this area of research in to the introduction, and methods*

340 *sections.*

*Authors Final Comments:*

*We have added a sentence to the introduction: Such algorithms are commonly categorised as "local" (e.g. PEST, Kim et al., 2007) or "global" (e.g. SCE, Duan et al., 1993), some examples of which have been compared by Wallner et al. (2012).*

**Reviewer Comment:**

345 2. Although the words "multi-objective optimization" aren't often written together in the text, this approach appears to be an a priori multi-objective optimization. By using the sum of each objective's rank as your objective, you have defined weightings a priori to merge multiple objectives into a single objective function. Please include at least one or two sentences explaining this and mentioning the difference between this and a posteriori multi-objective optimization (below).

*Authors Initial Comments:*

350 *Please see our response to point 3. below*

**Reviewer Comment:**

I mention this because you state that "multi-objective optimization methods have been advancing since the turn of the century", but this area has a pretty rich literature that goes back well into the 1990s. Additionally, most optimization researchers think of a posteriori (not a priori) when they think of multi-objective optimization. A posteriori approaches try to find a set of non-dominated Pareto optimal solutions and then select the best compromise afterwards. You might include references to other multi-objective papers that take this approach like:

"Multiobjective Automatic Parameter Calibration of a Hydrological Model" (Jung et al, 2017) "Comparing multi-objective optimization techniques to calibrate a conceptual hydrological model using in situ runoff and daily GRACE data" (Mostafaie et al. 2018) "Automatic calibration of HEC-HMS using single-objective and multi-objective PSO algorithms" (Kamali et al. 2013) "Multi-objective calibration of a distributed hydrological model (WetSpa) using a genetic algorithm" (Shafi and de Smedt 2009)

Or consider some of their references for older publications.

*Authors Initial Comments:*

*Yes we can see that this area of literature has been overlooked in the manuscript. We will add a few sentences on these approaches to the introduction.*

*Authors Final Comments:*

*We have added the sentences: "Multi-objective optimisation commonly involves seeking Pareto-optimal solutions that find a compromise between objective functions (e.g. Shafii and De Smedt, 2009; Kamali et al., 2013; Jung et al., 2017). Multi-objective methods may also be used to optimise more than one hydrological variable (e.g. Mostafaie et al., 2018)."*

**Reviewer Comment:**

3. Because of the a priori weighting (Comment #2), please provide information about how the multiple objectives are related to one another. Are some highly correlated? Negatively correlated? If, for instance, the rankings from the 4 high/water balance objectives operate as one and the 2 low flow indices operate as one, is there a concern that you are overweighting towards high flows?

*Authors Initial Comments:*

*We are not sure whether our approach would be considered a priori or a posteriori. Traditionally, a GLUE type approach would assign an a priori distribution to sample parameter values from, we chose to make no a priori assumptions and chose to sample from a uniform distribution across all 4 parameters. A GLUE approach would then weight the "behavioural" parameter sets a posteriori according to their metric scores. We have chosen 6 metrics, and have not "weighted" the runs by their scores, merely extracted the top 500. Yes, the 6 metrics we chose could be implicitly weighted according to their similarity (if two metrics were very similar, then they would hold more weight together than any one of the others). Thus, the graphic below demonstrates a quick look into their correlations. We have taken the LHS1, the "best run", for each catchment here, aside from the fact R is not easily capable of reading in 303 matrices of 500,000 rows, the sets of 500,000 for each catchment contain some truly awful parameterisations, and the metric interactions among these were understandably very odd. Here, we can generally see that there are significant correlations between each of the metrics: generally where catchments score well for one metric, they score well for all metrics. Bear in mind that for NSE and logNSE a high score is a good score, whilst for the other 4 a low score is a good score. The NSE and logNSE have the highest correlation, which would be expected, and the $MAM30_{APE}$ and $Q95_{APE}$ are also highly correlated as they are both errors in low flows. Correlations between MAPE and absPBIAS are positive with each other, and the low flows metrics, and negative with the NSE metrics. MAPE and $MAM30_{APE}$*

*are quite strongly correlated, as they are both mean percent error metrics. The metrics were carefully chosen to cover different elements of goodness of fit as follows:*

- *NSE – good at magnitude and timing of peak flows*
- *LogNSE – NSE on log flows in an attempt to match magnitude and timing of lower flows*
- *MAPE – overall magnitude of variability*

- *absPBIAS – total water balance*
- *$MAM30_{APE}$ – error in the lowest of flows*
- *$Q95_{APE}$ – fitting the tail of the FDC.*

*We would say that, if anything, these 6 metrics are together slightly more biased towards matching low flows than high flows, which we were happy with given their intended purpose for use in drought research.*

[Figure]

*Authors Final Comments:*

*We have significantly revised the section on rankings for clarity. We also added some sentences to the start of the calibration strategy section to place this approach in some context with existing literature. We hope that this, along with our explanation here, sufficiently resolves your query.*

**Reviewer Comment:**

4. Line 245-250: I find it surprising that there is a single very poor fit among nearly perfect fits, for example in Cornwall. As you are mentioning the reasons for poor fits in this paragraph, it is important to mention there does not appear to be a spatial pattern. Presumably, the same abstractions and groundwater issues affect the 0-10% threshold poor fit as its > 90% good fit neighbours. Are there any other feasible explanations?

*Authors Initial Comments:*

*We would argue that there IS a spatial pattern, there is generally good performance across the country, with the exception of two areas:*

- *some upland catchments in Scotland and Northern England that experience snowmelt contributions, and*

- *highly permeable catchments or those with significant human influence in south and south-eastern England. The more local scale variability across the south is likely due to the spatial variability in the geological units.*

*You have identified one exception to these two broad categories, which is the Warleggan in Cornwall. This catchment fails the thresholds due to the Nash Sutcliffe Efficiency metric alone: the peak flow magnitudes are significantly underestimated, the other metric scores are acceptable. This could be due to the fact that the catchment sits on a granite outcrop, so is less permeable than surrounding catchments, but the calibration process ought to be able to account for this; it would require further investigation to identify the cause of this specific insufficiency. We will add a comment about this exception to the manuscript.*

**Authors Final Comments:**

*The line was added: "For the Warleggan in Cornwall, poor performance is due to underestimation of peak flows, which may be attributed to an issue in simulating the localised geological outcrops."*

**Minor Comments**

**Reviewer Comment:**

Line 45 – Suggest 1 or 2 more references to fill out the discussion of low flow climate projections for the UK.

**Authors Initial Comments:**

*We have added Wilby and Harris (2006), Christierson et al (2012), and Prudhomme et al (2012)*

*Wilby, R. L., and Harris, I.: A framework for assessing uncertainties in climate change impacts: Low-flow scenarios for the River Thames, UK, Water Resour. Res., 42, W02419, 10.1029/2005wr004065, 2006. Christierson, B. v., Vidal, J.-P., and Wade, S. D.: Using UKCP09 probabilistic climate information for UK water resource planning, Journal of Hydrology, 424-425, 48-67, https://doi.org/10.1016/j.jhydrol.2011.12.020, 2012. Prudhomme, C., Young, A., Watts, G., Haxton, T., Crooks, S., Williamson, J., Davies, H., Dadson, S., and Allen, S.: The drying up of Britain? A national estimate of changes in seasonal river flows from 11 Regional Climate Model simulations, Hydrological Processes, 26, 1115-1118, 10.1002/hyp.8434, 2012.*

**Reviewer Comment:**

Line 70 – You may want to mention some proxy-based reconstructions; for example Jones et al (1984) "Riverflow reconstruction from tree rings in southern Britain" or the Old World Drought Atlas (Cook et al 2015) "Old World megadroughts and pluvials during the Common Era" which covers the UK.

**Authors Initial Comments:**

*We have added these references.*

**Reviewer Comment:**

Line 193 – Please define LHS500. This is the first time it is included in the text (only in the abstract).

**Authors Initial Comments:**

*The first reviewer also noticed this error, we have amended it to "The upper and lower daily limits of the 500 top ranking parameterisations (see Section **Error! Reference source not found.** for details on the ranking process) were used to calculate…"*

**Reviewer Comment:**

Table 2 – If possible, please try to fit the ranges on a single line of this table.

**Authors Initial Comments:**

*We've corrected this*

**Reviewer Comment:**

455    Lines 273 – You do a great job of describing a low UncW and low ContR as biased and under-sensitive - this is a helpful translation for readers. As a reader, I would also like a description of the converse. What does high UncW and high ContR mean?

*Authors Initial Comments:*

*We will add a sentence to this effect.*

460    **Reviewer Comment:**

Line 344 - Can you provide a description of which objective function(s) is driving the best fit parameter set in the Avon to consistently overestimate low flows?

*Authors Initial Comments:*

*We will look into the parameter values of the best run compared to the other LHS500 members, as well as the metric scores*
465    *to see if we can notice anything here.*

*Authors Final Comments:*

*Please see the histograms below which show the distributions of the model parameters, and objective functions for the LHS500, and the LHS1 as a black line for the Avon catchment. It looks as though the NSE scores are generally quite low for this catchment (<0.4). The threshold method will have prioritised any parameterisations that met the middle set of thresholds,*
470    *which will have pulled the few parameterisations that had an NSE of >0.4 to the top of the list. As the thresholds for MAM30 and Q95 were 75 for the middle thresholds, this has allowed a set with lower scores for these metrics than the average across the top 500, to become the "best" set. This means that the LHS1 for this catchment is actually slightly biased towards higher flows, and may explain the overestimation of low flows seen in the manuscript. Such trade-offs that are inherent in any multi-parameter optimisation make selecting a "best" parameterisation challenging, and this highlights the strength of utilising the*
475    *full set of 500 parameterisations.*

[Figure]

**Reviewer Comment:**

Line 372 – Please add the words "we consider" before "SSI values. . .". The thresholds of -1 and -1.5 are largely arbitrary and more of a convention than a true definition.

480    *Authors Initial Comments:*

*Valid point, we have added this to the manuscript*

**Reviewer Comment:**

Figure 9 – Please mention that you are plotting the mid-point of each event in the caption. It is currently only in the text (Line 417).

*Authors Initial Comments:*

*We have added this to the caption*

**Reviewer Comment:**

Figure 9 – For the Crimple watershed, there are 3 unique drought events for the Modeled data shown in the period 1975-1979. But Figure 8 shows only 2 crosses of the -1 threshold. Please confirm what is going on here.

*Authors Initial Comments:*

*The three modelled data circles suggest some discrepancy in the timing of the 1975/76 event among the LHS500, rather than 3 distinct events. The timing of the drought events is characterised by the dates the SSI crosses 0 (though we're only showing the events in Fig 9 where at least one month crosses SSI -1.5). The individual LHS500 runs demonstrate quite some width in the ascending limb as the SSI crosses into positive values in the Crimple in 1976/1977. This discrepancy in the end date of the drought event will affect its midpoint, and from Fig 9 it looks as though the LHS500 are grouped in to 3 main possibilities for timing. However, the thickest circle (demonstrating a higher number of runs) is the central one which best agrees with the timing of the observed event. The Greet and the Bush also show many circles for this event, and also have a wide band of grey LHS500 runs as the SSI crosses 0 in Fig 8. The Bush in particular doesn't cross back above SSI 0 until 1979 for some of the LHS 500 runs. We will re-read this section and make sure that it is clear that overlapping black circles suggest timing discrepancy rather than multiple events.*

*Authors Final Comments:*

[revised manuscript text omitted]